applied mathematics/statistics/mathematical modelling

stochastic differential equations, Markov chain Monte Carlo, Milstein scheme, parameter estimation, Bayesian data imputation

**Author for correspondence:**
Christiane Fuchs
e-mail: christiane.fuchs@uni-bielefeld.de

# Bayesian inference for diffusion processes: using higher-order approximations for transition densities

## Susanne Pieschner[1,2] and Christiane Fuchs[1,2,3]

[1]Institute of Computational Biology, Helmholtz Zentrum München, German Research Center for Environmental Health, Ingolstädter Landstr. 1, 85764 Neuherberg, Germany
[2]Department of Mathematics, Technische Universität München, Boltzmannstrasse 3, 85748 Garching, Germany
[3]Data Science Group, Faculty of Business Administration and Economics, Universität Bielefeld, Postfach 100131, 33501 Bielefeld, Germany

 SP, 0000-0002-2916-7782; CF, 0000-0003-3565-8315

Modelling random dynamical systems in continuous time, diffusion processes are a powerful tool in many areas of science. Model parameters can be estimated from time-discretely observed processes using Markov chain Monte Carlo (MCMC) methods that introduce auxiliary data. These methods typically approximate the transition densities of the process numerically, both for calculating the posterior densities and proposing auxiliary data. Here, the Euler–Maruyama scheme is the standard approximation technique. However, the MCMC method is computationally expensive. Using higher-order approximations may accelerate it, but the specific implementation and benefit remain unclear. Hence, we investigate the utilization and usefulness of higher-order approximations in the example of the Milstein scheme. Our study demonstrates that the MCMC methods based on the Milstein approximation yield good estimation results. However, they are computationally more expensive and can be applied to multidimensional processes only with impractical restrictions. Moreover, the combination of the Milstein approximation and the well-known modified bridge proposal introduces additional numerical challenges.

## 1. Introduction

Diffusion processes are used in many areas of science as a powerful tool to model continuous-time dynamical systems that are subject to random fluctuations. A diffusion process can be equivalently described by a stochastic differential equation (SDE).

If the SDE yields an analytical solution, the transition densities of the corresponding diffusion process are explicitly known and parameter estimation can be easily performed through a maximum likelihood approach, as demonstrated in [1]. However, in the majority of applications, this is not the case, and the transition densities are intractable.

When the transition densities are unknown, another challenge for parameter estimation is the type of available data. In practice, a process can only be observed at discrete points in time. A comprehensive overview of the methods for parameter inference from high-frequency data (i.e. where inter-observation times are small) can be found in [2, ch. 6]. For parameter estimation from low-frequency observations, Markov chain Monte Carlo (MCMC) techniques have been developed that introduce imputed data points to reduce the time steps between data points. This concept of Bayesian data imputation for the inference of diffusions has been used and developed further by many authors such as [3–6]. These methods are applicable to multidimensional processes and were extended for the case of latent process components as well as for the occurrence of measurement error. Thus, they are very promising for the use in real-data applications (e.g. [2,7]).

The concept of these MCMC algorithms is to construct a Markov chain whose elements are samples from the joint posterior density of the parameter and the imputed data points conditioned on the observations. This construction is achieved via a Gibbs sampling approach by alternately executing the following two steps:

(1) drawing the parameter conditional on the augmented path that consists of the observed data points and imputed data points and
(2) drawing the imputed data points conditional on the current parameter and the observed data points.

In both steps, direct sampling from the corresponding conditional distribution is generally not possible; therefore, a Metropolis–Hastings algorithm is applied. The (full conditional) posterior densities are reformulated as the product of the transition densities of the process in both steps and the prior density of the parameter in the first step. Because the transition densities are intractable, they can only be numerically approximated.

The numerical approximation of the transition densities of the process is necessary not only for calculating the posterior densities, but also for proposing the imputed data points. In both contexts, the Euler–Maruyama scheme is the standard approximation technique in the literature, including all of the aforementioned references. To reduce the amount of imputed data and the number of necessary iterations for the computationally expensive estimation method, one possible solution is to employ higher-order approximation schemes.

Therefore, we investigate the utilization and usefulness of such higher-order approximations on the example of the Milstein scheme. A closed form of the transition density based on the Milstein scheme is derived in [8]. In [9], this closed form is used to estimate the parameters of a hyperbolic diffusion process from high-frequency financial data, but not in the context of Bayesian data augmentation. For the latter, Elerian *et al*. [3] propose the possible use of the Milstein scheme. However, the specific implementation and benefit of this framework, in particular when using sophisticated proposal methods, remain unclear and, therefore, are the focus of this work. For our investigation, we first explain how to integrate the Milstein scheme into the framework of Bayesian data augmentation and then assess the effectiveness of this new combination in a simulation study which is a common approach in the literature (e.g. [10,11]).

This article is organized as follows. In §2, we define diffusion processes, describe the numerical approximation of their paths, and explain the derivation of the transition densities of the processes based on these approximations. In §3, we elaborate on the parameter estimation methods for diffusion processes using Bayesian data augmentation and the approximated transition densities. In §4, we give some comments about our implementation of these methods and in §5, we explain the set-up of our simulation study. In §§6 and 7, we present the results and discussion. The source code of our implementation and the simulation study is publicly available at https://github.com/fuchslab/Inference_for_SDEs_with_the_Milstein_scheme.

## 2. Approximation of the transition density of a diffusion process

We consider a $d$-dimensional *time-homogeneous Itô diffusion process*, $(X_t)_{t \geq 0}$, a stochastic process that fulfils the following SDE:

$$dX_t = \mu(X_t, \theta)\, dt + \sigma(X_t, \theta)\, dB_t, \quad X_0 = x_0, \tag{2.1}$$

with state space $\mathcal{X} \subseteq \mathbb{R}^d$, starting value $x_0 \in \mathcal{X}$, and a $q$-dimensional Brownian motion, $(B_t)_{t \geq 0}$. The model parameter $\theta \in \Theta$ is from an open set $\Theta \subseteq \mathbb{R}^p$. In addition, we assume that the drift function $\mu: \mathcal{X} \times \Theta \to \mathbb{R}^d$ and diffusion function $\sigma: \mathcal{X} \times \Theta \to \mathbb{R}^{d \times q}$ fulfil the Lipschitz condition and growth bound to ensure that (2.1) has a unique solution (e.g. [12, ch. 5]).

In this work, we use rather simple, well-known examples of such a diffusion process in order to focus on the investigated estimation methods and make the article easy to follow. Our example for the main text is the geometric Brownian motion (GBM), and in appendix D, we also provide all relevant details for the Cox–Ingersoll–Ross (CIR) process. The GBM is described by the following SDE:

$$dX_t = \alpha X_t \, dt + \sigma X_t \, dB_t, \quad X_0 = x_0, \tag{2.2}$$

with state space $\mathcal{X} = \mathbb{R}_+$, starting value $x_0 \in \mathcal{X}$ and the two-dimensional parameter $\theta = (\alpha, \sigma)^T$, where $\alpha \in \mathbb{R}$ and $\sigma \in \mathbb{R}_+$, $\mathbb{R}_+$ being the set of all strictly positive real numbers. The GBM is especially suitable as a benchmark model because it has an explicit solution. The stochastic process

$$X_t = x_0 \exp\left( \left( \alpha - \frac{1}{2}\sigma^2 \right)t + \sigma B_t \right),$$

fulfils (2.2) for all $t \geq 0$. Hence, the multiplicative increments of the GBM are lognormally distributed as follows:

$$\frac{X_t}{X_s} \sim \mathcal{LN}\left( \left( \alpha - \frac{1}{2}\sigma^2 \right)(t-s), \sigma^2(t-s) \right),$$

for $t \geq s \geq 0$, and the transition density is explicitly known as

$$p(s, x, t, y) = P(X_t = y \mid X_s = x)$$
$$= \frac{1}{\sqrt{2\pi(t-s)}\sigma y} \exp\left( -\frac{(\log y - \log x - (\alpha - \frac{1}{2}\sigma^2)(t-s))^2}{2\sigma^2(t-s)} \right). \tag{2.3}$$

A derivation of the solution of the GBM and its transition density can be found in [13].

In different contexts, one often considers the logarithm of the GBM, $\log X_t$, which is simply a normally distributed random variable for fixed $t$, with corresponding SDE

$$d(\log X_t) = \left( \alpha - \frac{1}{2}\sigma^2 \right)dt + \sigma \, dB_t, \quad \log X_0 = \log x_0. \tag{2.4}$$

However, we do not employ this transformation here because of the constant diffusion function in (2.4). For the log-transformed GBM, the approximation methods that we wish to compare would yield an identical approximation.

## 2.1. Approximation of the solution of an SDE

Unlike the GBM, most SDEs do not have an analytical solution; thus, their transition densities are not explicitly known. Instead, numerical approximation schemes are used for the solution of the SDEs. Kloeden & Platen [14] have provided a detailed description of these methods. Several of the approximation schemes are based on the stochastic Taylor expansion. For a general treatment of this expansion, we refer the interested reader to [14]. The most commonly used approximation is the Euler(–Maruyama) scheme, which approximates the $d$-dimensional solution $(X_t)_{t \geq 0}$ of an SDE by setting $Y_0 = x_0$ and, then, successively calculating the following:

$$Y_{k+1} = Y_k + \mu(Y_k, \theta)\Delta t_k + \sigma(Y_k, \theta)\Delta B_k, \tag{2.5}$$

where $\Delta t_k = t_{k+1} - t_k$, $\Delta B_k = B_{t_{k+1}} - B_{t_k}$ and $Y_k$ is the approximation of $X_{t_k}$ for $k = 0, 1, 2, \ldots$. The approximation improves as the time step $\Delta t_k$ decreases. The Euler scheme contains only the time component and the stochastic integral of multiplicity one from the stochastic Taylor expansion of process $(X_t)_{t \geq 0}$, and has strong order of convergence 0.5.

A discrete-time approximation $Y^\Delta$ with maximum step size $\Delta > 0$ converges with strong order $\gamma > 0$ at time $T$ to the solution $X_T$ of a given SDE if there exists a positive constant $C$ independent of $\Delta$ and a $\Delta_0 > 0$ such that

$$E(|X_T - Y_T^\Delta|) \leq C\Delta^\gamma,$$

for all $\Delta \in (0, \Delta_0)$. Strong convergence ensures a pathwise approximation of the solution process $(X_t)_{t \geq 0}$ of the given SDE. The higher the order of strong convergence is, the faster the mean absolute error between the approximation and the solution decreases as the maximum time step size $\Delta$ decreases.

By adding another term of the stochastic Taylor expansion to equation (2.5), one obtains the Milstein scheme that approximates the $d$-dimensional process $(X_t)_{t \geq 0}$ by setting $Y_0 = x_0$ and, then, successively calculating for the $i$th component

$$
\begin{aligned}
Y_{k+1}^{(i)} = Y_k^{(i)} + \mu_i(Y_k, \theta)\Delta t_k + \sum_{j=1}^{q} \sigma_{ij}(Y_k, \theta)\Delta B_k^{(j)} \\
+ \sum_{j=1}^{q}\sum_{l=1}^{q}\sum_{r=1}^{d} \sigma_{rj}(Y_k, \theta)\frac{\partial \sigma_{il}}{\partial y^{(r)}}(Y_k, \theta)\int_{t_k}^{t_{k+1}}\int_{t_k}^{s} \mathrm{d}B_u^{(j)}\mathrm{d}B_s^{(l)}
\end{aligned}
\tag{2.6}
$$

for $k = 0, 1, \ldots$ and $i = 1, \ldots, d$.

When $\sigma(Y_k, \theta)$ is constant in $Y_k$, the last term vanishes and the Milstein scheme reduces to the Euler scheme. If $\mu(Y_k, \theta)$ is once continuously differentiable and $\sigma(Y_k, \theta)$ is twice continuously differentiable regarding $Y_k$, then, the Milstein scheme is strongly convergent of order 1.0, which is higher than that of the Euler scheme. An illustration of this difference in the simulation of SDE trajectories is presented e.g. in [15]. However, there is a severe restriction on the practical applicability of the Milstein scheme because the stochastic double integral in the last term of (2.6) only yields an analytical solution for $j = l$. Although approximation techniques for the double integral exist (e.g. [14]), they are unsuitable for our purposes. On the one hand, we wish to avoid adding yet another layer of approximation and, thus, additional computational time. On the other hand, we must find the distribution of $Y_{k+1}$ based on approximation schemes (2.5) and (2.6), which is also not explicitly possible when adding another approximation. For this reason, we focus on models where the double integral appears exclusively for the same components of the Brownian motion. For example, this is the case when the process is driven by a one-dimensional Brownian motion (i.e. the diffusion function $\sigma(Y_k, \theta)$ is of dimension $d \times 1$). Hence, the diffusion model includes only one source of noise that may affect each of the components of the process. More generally, we require that

$$
\sigma_{rj}(Y_k, \theta)\frac{\partial \sigma_{il}}{\partial y^{(r)}}(Y_k, \theta) \equiv 0 \quad \text{for } j \neq l,
\tag{2.7}
$$

so that only $j = l$ is inside the double integral. Relation (2.7) implies the following:

— if an entry $\sigma_{rj}(Y_k, \theta)$ is non-zero, then the entries of all *other* columns and *all* rows must not depend on $Y_k^{(r)}$, and
— if an entry $\sigma_{il}(Y_k, \theta)$ depends on $Y_k^{(r)}$, then the entries of all *other* columns in row $r$ must be zero.

In particular, this means that unless the $r$th row of the diffusion function contains only zeros, component $Y_k^{(r)}$ can only appear in *one* column of the diffusion function (and if it appears, then the entries of all *other* columns in row $r$ must be zero). Moreover, each component of the diffusion process $(X_t)_{t \geq 0}$ can only be directly affected by more than one component of the Brownian motion, if the size of all stochastic effects (i.e. *all* entries of the diffusion function) does not depend on the respective component of the diffusion process. Further, if all $d$ components of the diffusion process appear in the diffusion function, then the process can be affected by at most $d$ components of the Brownian motion. Besides, if all $d$ components of the diffusion process appear in the diffusion function and the process shall be affected by $d$ components of the Brownian motion, the diffusion function must be a (possibly column-wise permuted) diagonal matrix. In many applications, these are not realistic assumptions.

Assume that the $i$th component of the diffusion process appears in the $i$th row of the diffusion function and that the respective entry of the diffusion function does not depend on the remaining components $Y_k^{(r)}$, $r \neq i$ (the contrary would impose restrictions on other rows, as described above). Then, the $i$th component of the approximated process is

$$
\begin{aligned}
Y_{k+1}^{(i)} = Y_k^{(i)} + \mu_i(Y_k, \theta)\Delta t_k + \sigma_{ij}(Y_k, \theta)\Delta B_k^{(j)} \\
+ \sigma_{ij}(Y_k, \theta)\frac{\partial \sigma_{ij}}{\partial y^{(i)}}(Y_k, \theta)\frac{1}{2}((\Delta B_k^{(j)})^2 - \Delta t_k)
\end{aligned}
\tag{2.8}
$$

for $k = 0, 1, \ldots$ and where $j$ is the column index of the one non-zero entry depending on $Y_k^{(i)}$ in the $i$th row of the diffusion function.

Moreover, note that if we consider the approximation $Y_{k+1}^{(i)}$ in equation (2.8) as a function $g(\Delta B_k^{(j)})$ of the increment of the Brownian motion, $g$ is quadratic in $\Delta B_k^{(j)}$. Therefore, the function $g$ has a global extremum with value

$$g^* = Y_k^{(i)} - \frac{1}{2}\sigma_{ij}(Y_k, \theta) \Big/ \left(\frac{\partial \sigma_{ij}}{\partial y^{(i)}}(Y_k, \theta)\right) + \left(\mu_i(Y_k, \theta) - \frac{1}{2}\sigma_{ij}(Y_k, \theta)\frac{\partial \sigma_{ij}}{\partial y^{(i)}}(Y_k, \theta)\right)\Delta t_k. \tag{2.9}$$

Hence, there is a bound on the range of possible values for $Y_{k+1}^{(i)}$ resulting from the Milstein scheme which might exclude values that the solution process $X_{t_k}$ could take. Whether this is a lower or upper bound depends on the sign of the diffusion function and its derivative. The second derivative of $g$ is given by

$$\frac{\partial^2 g(\Delta B_k^{(j)})}{\partial(\Delta B_k^{(j)})^2} = \sigma_{ij}(Y_k, \theta)\frac{\partial \sigma_{ij}}{\partial y^{(i)}}(Y_k, \theta) =: g''.$$

Thus, the extremum $g^*$ is a maximum and puts an upper bound on the possible values of $Y_{k+1}^{(i)}$ if $g'' < 0$, and $g^*$ is a minimum and puts a lower bound on $Y_{k+1}^{(i)}$ if $g'' > 0$. For the case where $g'' = 0$, the Milstein scheme reduces to the Euler scheme.

Since our example, the GBM, is a one-dimensional process, the double integral in equation (2.6) vanishes and the Milstein scheme for the GBM yields the following:

$$Y_{k+1} = Y_k + \alpha Y_k \Delta t_k + \sigma Y_k \Delta B_k + \frac{1}{2}\sigma^2 Y_k((\Delta B_k)^2 - \Delta t_k),$$

for $k = 0, 1, \ldots$, where the first three summands also correspond to the Euler scheme. Figure 1 illustrates the two approximation schemes. It presents three trajectories of the GBM, which are represented by red points and which were simulated by setting a seed for the random number generator and, then, sampling from the explicit transition density (2.3). The same seed was used to sample the increments of the Brownian motion from the normal density and then transform them by (2.5) and (2.6) to obtain the Euler (black) and the Milstein (blue) approximation of the trajectories. We observe that in almost all cases, the Milstein approximation is either closer to or as close to the points of the trajectories as the Euler approximation.

## 2.2. Transition densities based on approximation schemes

While sampling diffusion paths is fairly straightforward for both approximation schemes as described above, determining the corresponding transition density is less apparent for the Milstein scheme. Since the Euler scheme is a linear transformation of $\Delta B_k \sim \mathcal{N}(0, \sqrt{\Delta t_k}I_q)$, where $I_q$ denotes the $m$-dimensional identity matrix, the transition density derived from the Euler scheme is also a multivariate Gaussian density:

$$\pi^{\text{Euler}}(Y_{k+1}|Y_k) = \phi(Y_{k+1}|Y_k + \mu(Y_k, \theta)\Delta t_k, \sigma(Y_k, \theta)\sigma^{\mathrm{T}}(Y_k, \theta)\Delta t_k),$$

where $\phi(y\,|\,a, b)$ denotes the multivariate Gaussian density with mean $a \in \mathbb{R}^d$ and covariance matrix $b \in \mathbb{R}^{d \times d}$ evaluated at $y$.

For the Milstein scheme, deriving the transition density is more complicated, even in the case of a one-dimensional diffusion process, which we consider here. Elerian [8] derived the transition density by first rearranging the Milstein scheme to obtain a transformation of a non-central chi-squared distributed variable for which the density is known, and then applying the random variable transformation theorem. In appendix A, we present an alternative derivation that directly applies the random variable transformation theorem to $\Delta B_k$. Both approaches produce the same result. For simplicity of notation, we set $\mu_k := \mu(Y_k, \theta)$, $\sigma_k := \sigma(Y_k, \theta)$ and $\sigma_k' := \partial \sigma(y, \theta)/\partial y\,|_{y=Y_k}$. Then, the transition density based on the Milstein approximation for a one-dimensional diffusion process is as follows:

$$\pi^{Mil}(Y_{k+1}|Y_k) = \frac{\exp\left(\frac{-C_k(Y_{k+1})}{D_k}\right)}{\sqrt{2\pi}\sqrt{\Delta t_k}\sqrt{A_k(Y_{k+1})}} \cdot \left[\exp\left(-\frac{\sqrt{A_k(Y_{k+1})}}{D_k}\right) + \exp\left(\frac{\sqrt{A_k(Y_{k+1})}}{D_k}\right)\right]$$

with

$$A_k(Y_{k+1}) = (\sigma_k)^2 + 2\sigma_k \sigma_k'\left(Y_{k+1} - Y_k - \left(\mu_k - \frac{1}{2}\sigma_k \sigma_k'\right)\Delta t_k\right),$$

$$C_k(Y_{k+1}) = \sigma_k + \sigma_k'\left(Y_{k+1} - Y_k - \left(\mu_k - \frac{1}{2}\sigma_k \sigma_k'\right)\Delta t_k\right),$$

$$D_k = \sigma_k(\sigma_k')^2 \Delta t_k$$

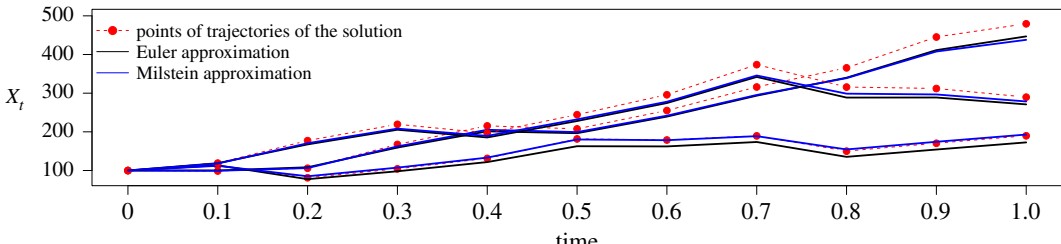

**Figure 1.** Three trajectories of a GBM (2.2) with $\alpha = 1$ and $\sigma^2 = 0.25$ and their approximations by the Euler and the Milstein scheme.

and for

$$Y_{k+1} \geq Y_k - \frac{1}{2}\frac{\sigma_k}{\sigma'_k} + \left(\mu_k - \frac{1}{2}\sigma_k\sigma'_k\right)\Delta t_k, \quad \text{if } \sigma_k\sigma'_k > 0 \tag{2.10}$$

and

$$Y_{k+1} \leq Y_k - \frac{1}{2}\frac{\sigma_k}{\sigma'_k} + \left(\mu_k - \frac{1}{2}\sigma_k\sigma'_k\right)\Delta t_k, \quad \text{if } \sigma_k\sigma'_k < 0. \tag{2.11}$$

The bounds in (2.10) and (2.11) coincide with the bound (2.9) on the range of possible values $Y_{k+1}$ resulting from the Milstein scheme in §2.1. For values of $Y_{k+1}$ within the respective bound, $A_k(Y_{k+1})$ is non-negative and its square root takes real values; otherwise, the transition density is equal to zero. Hence, there is a lower or an upper bound on the support of $\pi^{Mil}$. Moreover, one can show that the value of the transition density tends to infinity as $Y_{k+1}$ approaches the bound. However, the interval for which the density increases towards infinity may be arbitrarily narrow depending on the parameter setting.

For the GBM, we have $\sigma(X_t, \theta) = \sigma X_t$ with parameter $\sigma > 0$, the process taking values in $\mathbb{R}_+$. Therefore, we obtain a lower bound for the possible values of $Y_{k+1}$

$$Y_{k+1} \geq Y_k\left(\frac{1}{2} + \left(\alpha - \frac{1}{2}\sigma^2\right)\Delta t_k\right). \tag{2.12}$$

Depending on the parameter combination $\theta = (\alpha, \sigma)^T$, this lower bound may be negative, in which case the support of the transition density includes the entire state space of the GBM.

In figure 2, we illustrate the transition densities based on the GBM solution, Euler scheme and Milstein scheme for two different parameter settings. We observe that the Milstein transition density better approximates the mode of the transition density of the solution than the Euler transition density does. On the other hand, while the support of the Euler transition density is the set of all real numbers, the Milstein transition density puts zero weight on the values of $Y_{k+1}$ that are below the lower bound, even though some of the values are feasible according to the transition density of the solution process.

Other approximation methods for the transition densities were developed for example in [16–18]. Here, we focus on the numerical approximation methods described above. Because for the estimation methods introduced in the next section, it is crucial to not only be able to approximate the transition density, but also sampling from the resulting density needs to be possible and fast.

## 3. Bayesian data augmentation for the parameter estimation of diffusions

With low-frequency observations $X^{\text{obs}} = (X_{\tau_0}, \ldots, X_{\tau_M})$ of the process $(X_t)_{t\geq 0}$ described by the SDE (2.1), we wish to estimate parameter $\theta$. In this work, we assume that all observations are complete (i.e. there are no latent or unobserved components for all observations) and that there are no measurement errors. The approximation schemes for the solution of the SDE as introduced in §2 are only appropriate for small time steps. Therefore, we introduce additional data points $X^{\text{imp}}$ at intermediate time points (as visualized in figure 3 and explained in detail in §3.2) and estimate the parameter $\theta$ from the augmented path $\{X^{\text{obs}}, X^{\text{imp}}\}$. To this end, a two-step MCMC approach is used to construct the Markov chain $\{\theta_{(i)}, X_{(i)}^{\text{imp}}\}_{i=1,\ldots,L}$, the elements of which are samples from the joint posterior distribution $\pi(\theta, X^{\text{imp}} \mid X^{\text{obs}})$:

Step (1) *Parameter update*: Draw $\theta_{(i)} \sim \pi(\theta_{(i)} \mid X^{\text{obs}}, X_{(i-1)}^{\text{imp}})$,
Step (2) *Path update*: Draw $X_{(i)}^{\text{imp}} \sim \pi(X_{(i)}^{\text{imp}} \mid X^{\text{obs}}, \theta_{(i)})$.

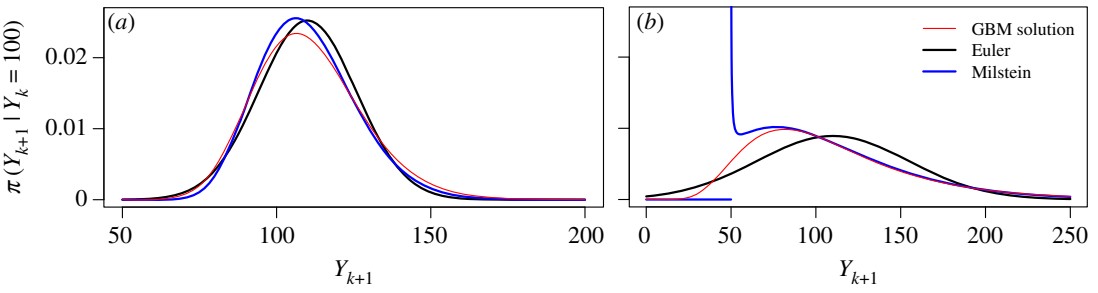

**Figure 2.** Transition densities for a transition from $Y_k$ to $Y_{k+1}$ with a time step of $\Delta t_k = 0.1$ for two different parameter settings based on the GBM solution, Euler scheme and Miltstein scheme, respectively. (a) $\alpha = 1$, $\sigma^2 = 0.25$ and $Y_k = 100$, (b) $\alpha = 1$, $\sigma^2 = 2$ and $Y_k = 100$.

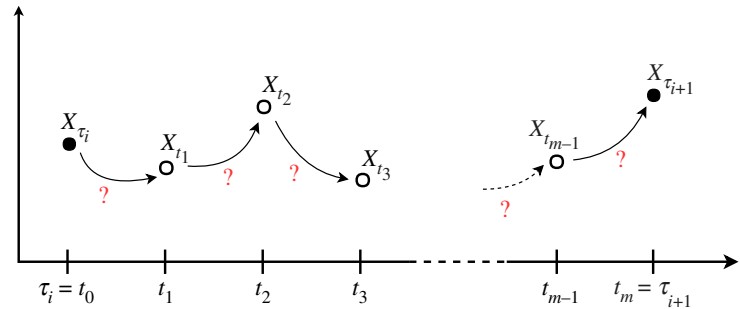

**Figure 3.** Augmented path segment: filled circles represent observed data points and open circles represent imputed points.

A general introduction to MCMC methods is presented in [19]. The resulting MCMC chain $\{\theta_{(i)}, X^{\mathrm{imp}}_{(i)}\}_{i=l+1,\dots,L}$, after discarding the first $l$ elements as burn-in, can be considered a sample drawn from the joint posterior distribution $\pi(\theta, X^{\mathrm{imp}} \mid X^{\mathrm{obs}})$ and can be used for a fully Bayesian analysis. The two steps of the algorithm are described in detail in the following two subsections. We use $\pi$ to denote the exact densities of the process that is the (full conditional) posterior densities as well as the transition densities. The meaning becomes clear from the arguments. Approximated densities are indicated by a corresponding superscript.

## 3.1. Parameter update

In Step (1), a parameter proposal $\theta^*$ is drawn from a proposal density $q(\theta^* \mid \theta, X^{\mathrm{obs}}, X^{\mathrm{imp}})$ which may or may not depend on the imputed and observed data. If a proposal $\theta^* = \theta + u$ with an update $u$ that is independent of the current parameter value $\theta$ is used, the proposal strategy is called a random walk proposal. Proposal $\theta^*$ is accepted with the following probability:

$$\zeta(\theta^*, \theta) = 1 \wedge \frac{\pi(\theta^* \mid X^{\mathrm{obs}}, X^{\mathrm{imp}})\, q(\theta \mid \theta^*, X^{\mathrm{obs}}, X^{\mathrm{imp}})}{\pi(\theta \mid X^{\mathrm{obs}}, X^{\mathrm{imp}})\, q(\theta^* \mid \theta, X^{\mathrm{obs}}, X^{\mathrm{imp}})}.$$

Otherwise, the previous $\theta$ value is kept.

Due to Bayes' theorem and the fact that a diffusion process has the Markov property, the (full conditional) posterior density can be represented as

$$\pi(\theta \mid X^{\mathrm{obs}}, X^{\mathrm{imp}}) \propto \left( \prod_{k=0}^{n-1} \pi(X_{t_{k+1}} \mid X_{t_k}, \theta) \right) p(\theta),$$

where $\pi(X_{t_{k+1}} \mid X_{t_k}, \theta)$ denotes the transition density of the process $(X_t)_{t \geq 0}$, $n+1$ is the total number of data points in the augmented path, and $p$ denotes the prior density of the parameter. We choose a random walk proposal where the $r$ components of $\theta^*$ that take values on the entire real line $\mathbb{R}$ are drawn from

the normal distribution $\mathcal{N}(\theta_j, \gamma_j^2)$ for $j = 1, \ldots, r$ and some predefined $\gamma_j \in \mathbb{R}_+$. The (remaining) strictly positive components are drawn from a lognormal distribution $\mathcal{LN}(\log \theta_j, \gamma_j^2)$, for $j = r + 1, \ldots, p$. In this case, the acceptance probability reduces to

$$\zeta(\theta^*, \theta) = 1 \wedge \left( \prod_{k=0}^{n-1} \frac{\pi(X_{t_{k+1}} \mid X_{t_k}, \theta^*)}{\pi(X_{t_{k+1}} \mid X_{t_k}, \theta)} \right) \frac{p(\theta^*)}{p(\theta)} \left( \prod_{j=r+1}^{p} \frac{\theta_j^*}{\theta_j} \right), \tag{3.1}$$

as derived in [2, ch. 7.1.3].

The transition density $\pi(X_{t_{k+1}} \mid X_{t_k}, \theta)$ is generally not explicitly known, but it can be approximated by the Euler or Milstein scheme as described in §2.

## 3.2. Path update

Since a diffusion process has the Markov property, the likelihood function of parameter $\theta$ factorizes as

$$\pi(X_{\tau_0}, \ldots, X_{\tau_M} \mid \theta) = \pi(X_{\tau_0} \mid \theta) \prod_{i=1}^{M} \pi(X_{\tau_i} \mid X_{\tau_{i-1}}, \theta), \tag{3.2}$$

and the latent path segments between observations are conditionally independent given the observations. Hence, it is sufficient to consider the imputation problem in Step (2) only for one path segment between two consecutive observations $X_{\tau_i}$ and $X_{\tau_{i+1}}$. As figure 3 illustrates, the time interval between the two observations is divided into $m$ subintervals, such that the endpoints of these intervals are $\tau_i = t_0 < t_1 < \ldots < t_m = \tau_{i+1}$ and the time steps are $\Delta t_k = t_{k+1} - t_k$ for $k = 0, \ldots, m - 1$. We denote the observations by $X_{\{\tau_i, \tau_{i+1}\}}^{\text{obs}} = \{X_{\tau_i}, X_{\tau_{i+1}}\}$ and the imputed data points by $X_{(\tau_i, \tau_{i+1})}^{\text{imp}} = \{X_{t_1}, \ldots, X_{t_{m-1}}\}$.

After initializing the imputed data by linear interpolation, the path is updated using the Metropolis–Hastings algorithm. A proposal $X_{(\tau_i, \tau_{i+1})}^{\text{imp}*}$ is drawn from a distribution with density $q$, which may depend on the observed data, current imputed data and parameter $\theta$. It is accepted with the following probability:

$$\zeta(X_{(\tau_i, \tau_{i+1})}^{\text{imp}*}, X_{(\tau_i, \tau_{i+1})}^{\text{imp}}) = 1 \wedge \frac{\pi(X_{(\tau_i, \tau_{i+1})}^{\text{imp}*} \mid X_{\{\tau_i, \tau_{i+1}\}}^{\text{obs}}, \theta) \, q(X_{(\tau_i, \tau_{i+1})}^{\text{imp}} \mid X_{(\tau_i, \tau_{i+1})}^{\text{imp}*}, X_{\{\tau_i, \tau_{i+1}\}}^{\text{obs}}, \theta)}{\pi(X_{(\tau_i, \tau_{i+1})}^{\text{imp}} \mid X_{\{\tau_i, \tau_{i+1}\}}^{\text{obs}}, \theta) \, q(X_{(\tau_i, \tau_{i+1})}^{\text{imp}*} \mid X_{(\tau_i, \tau_{i+1})}^{\text{imp}}, X_{\{\tau_i, \tau_{i+1}\}}^{\text{obs}}, \theta)}. \tag{3.3}$$

Otherwise, the proposal is discarded and the previously imputed data $X_{(\tau_i, \tau_{i+1})}^{\text{imp}}$ is kept. Due to the Markov property, we have:

$$\frac{\pi(X_{(\tau_i, \tau_{i+1})}^{\text{imp}*} \mid X_{\{\tau_i, \tau_{i+1}\}}^{\text{obs}}, \theta)}{\pi(X_{(\tau_i, \tau_{i+1})}^{\text{imp}} \mid X_{\{\tau_i, \tau_{i+1}\}}^{\text{obs}}, \theta)} = \prod_{k=0}^{m-1} \frac{\pi(X_{t_{k+1}}^* \mid X_{t_k}^*, \theta)}{\pi(X_{t_{k+1}} \mid X_{t_k}, \theta)},$$

where $X_{t_0}^* = X_{t_0} = X_{\tau_i}$, $X_{t_m}^* = X_{t_m} = X_{\tau_{i+1}}$ and $\pi(X_{t_{k+1}} \mid X_{t_k}, \theta)$ denotes the transition density of process $(X_t)_{t \geq 0}$.

The challenging aspect of the path update step involves determining how to propose new points. The simplest approach uses the (approximated) transition density to propose a new point by conditioning only on the point to the left of the new point. We call this proposal method the *left-conditioned proposal* and illustrate it in figure 4a. The proposal density of an entire path segment is simply the product

$$q_{LC}(X_{(\tau_i, \tau_{i+1})}^{\text{imp}*} \mid X_{\tau_i}, \theta) = \prod_{k=0}^{m-2} \pi(X_{t_{k+1}}^* \mid X_{t_k}^*, \theta),$$

where $X_{t_0}^* = X_{\tau_i}$. Thus, the acceptance probability reduces to

$$\zeta(X_{(\tau_i, \tau_{i+1})}^{\text{imp}*}, X_{(\tau_i, \tau_{i+1})}^{\text{imp}}) = 1 \wedge \left( \prod_{k=0}^{m-1} \frac{\pi(X_{t_{k+1}}^* \mid X_{t_k}^*, \theta)}{\pi(X_{t_{k+1}} \mid X_{t_k}, \theta)} \right) \left( \prod_{k=0}^{m-2} \frac{\pi(X_{t_{k+1}} \mid X_{t_k}, \theta)}{\pi(X_{t_{k+1}}^* \mid X_{t_k}^*, \theta)} \right)$$

$$= 1 \wedge \frac{\pi(X_{\tau_{i+1}} \mid X_{t_{m-1}}^*, \theta)}{\pi(X_{\tau_{i+1}} \mid X_{t_{m-1}}, \theta)},$$

where $X_{t_m}^* = X_{t_m} = X_{\tau_{i+1}}$. Here, the transition density can again be approximated by the Euler or Milstein scheme from §2.

This proposal strategy considers the information from the observation $X_{\tau_i}$ on the left, while the proposed path segment is independent of the observation $X_{\tau_{i+1}}$ on the right. This may lead to a large

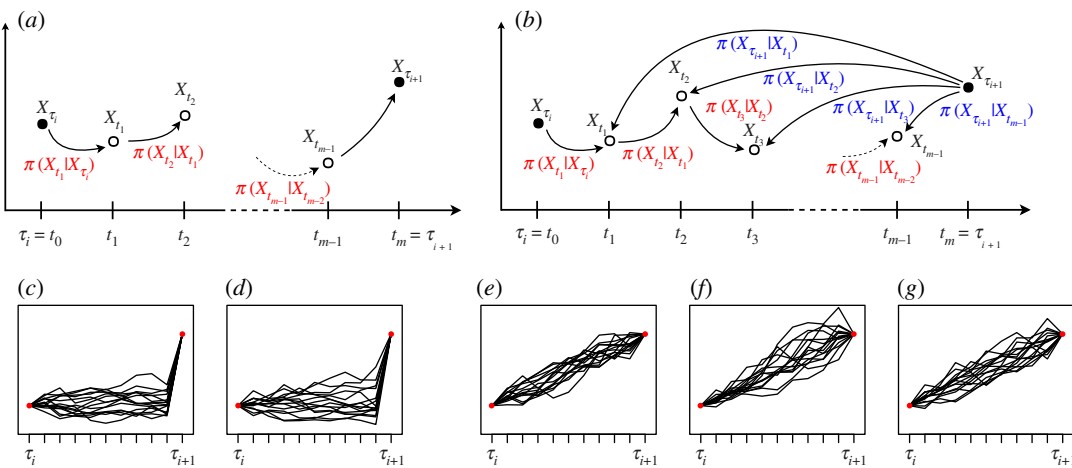

**Figure 4.** Different proposal strategies (a)–(b) and realizations using different approximation schemes (c)–(g). (a) Left-conditioned proposal, (b) bridge proposal, (c) left-conditioned Euler proposal, (d) left-conditioned Milstein proposal, (e) modified bridge Euler proposal, (f) modified bridge Milstein proposal, (g) diffusion bridge Milstein proposal.

jump in the last step from $X_{t_{m-1}}$ to $X_{\tau_{i+1}}$, as can be seen in figure 4c,d, and hence, to an improbable transition. Therefore, the acceptance probability for the left-conditioned proposal $X^{\mathrm{imp}*}_{(\tau_i, \tau_{i+1})}$, and consequently, the acceptance rate of the MCMC sampler is usually low.

A number of more sophisticated proposal strategies have been suggested. Ch. 7.1 in [2] reviews some of these. Here, we first consider the *modified bridge (MB) proposal*, which conditions on both the previous data point and the following observation on the right, as visualized in figure 4b. This strategy was originally proposed by Durham & Gallant [20] and first applied in the Bayesian framework in [21]. More recently, Whitaker *et al.* [10] suggested improved bridge constructs, and van der Meulen & Schauer [22] proposed the so-called guided proposals.

For the MB proposal, the proposal density of an entire path segment factorizes again as follows:

$$q_{MB}(X^{\mathrm{imp}*}_{(\tau_i, \tau_{i+1})} \mid X_{\tau_i}, X_{\tau_{i+1}}, \theta) = \prod_{k=0}^{m-2} \pi(X^*_{t_{k+1}} \mid X^*_{t_k}, X_{\tau_{i+1}}, \theta),$$

where $X^*_{t_0} = X_{\tau_i}$. We apply Bayes' theorem and the Markov property to rewrite the left- and right-conditioned proposal density of one point as

$$\pi(X^*_{t_{k+1}} \mid X^*_{t_k}, X_{\tau_{i+1}}, \theta) \propto \pi(X^*_{t_{k+1}} \mid X^*_{t_k}, \theta)\, \pi(X_{\tau_{i+1}} \mid X^*_{t_{k+1}}, \theta), \tag{3.4}$$

for $k = 0, \ldots, m-2$.

In [20], it is suggested to approximate the two transition densities on the right-hand side by the Euler scheme and to further approximate $\mu(X^*_{t_{k+1}}, \theta)$ and $\sigma(X^*_{t_{k+1}}, \theta)$ by $\mu(X^*_{t_k}, \theta)$ and $\sigma(X^*_{t_k}, \theta)$, respectively. This way, they obtain that (3.4) is approximately proportional to a Gaussian density which we will use for the MB proposal based on the Euler scheme

$$\pi^{\mathrm{Euler}}(X^*_{t_{k+1}} \mid X^*_{t_k}, X_{\tau_{i+1}}, \theta)$$
$$= \phi\left(X^*_{t_{k+1}} \,\middle|\, X^*_{t_k} + \left(\frac{X_{\tau_{i+1}} - X^*_{t_k}}{\tau_{i+1} - t_k}\right)\Delta t_k, \left(\frac{\tau_{i+1} - t_{k+1}}{\tau_{i+1} - t_k}\right)\Sigma(X^*_{t_k}, \theta)\Delta t_k\right), \tag{3.5}$$

where $\Sigma(X^*_{t_k}, \theta) = \sigma^2(X^*_{t_k}, \theta)$ and $\phi$ is defined in §2.2.

We now consider the Milstein approximation for the two factors on the right-hand side of (3.4). The first factor resembles the Milstein transition density stated in §2.2. With the same notation, $\Delta_+ = t_m - t_{k+1}$, and $t_m = \tau_{i+1}$, the second factor is as follows:

$$\pi^{Mil}(X_{t_m} \mid X^*_{t_{k+1}}, \theta) = \frac{\exp\left(-F_m(X^*_{t_{k+1}})/G_m(X^*_{t_{k+1}})\right)}{\sqrt{2\pi}\sqrt{\Delta_+}\sqrt{E_m(X^*_{t_{k+1}})}}$$

$$\times \left[\exp\left(-\frac{\sqrt{E_m(X^*_{t_{k+1}})}}{G_m(X^*_{t_{k+1}})}\right) + \exp\left(\frac{\sqrt{E_m(X^*_{t_{k+1}})}}{G_m(X^*_{t_{k+1}})}\right)\right]$$

with

$$E_m(X_{t_{k+1}}^*) = (\sigma_{k+1}^*)^2 + 2\sigma_{k+1}^* \sigma'^*_{k+1}\left(X_{t_m} - X_{t_{k+1}}^* - \left(\mu_{k+1}^* - \frac{1}{2}\sigma_{k+1}^*\sigma'^*_{k+1}\right)\Delta_+\right),$$

$$F_m(X_{t_{k+1}}^*) = \sigma_{k+1}^* + \sigma'^*_{k+1}\left(X_{t_m} - X_{t_{k+1}}^* - \left(\mu_{k+1}^* - \frac{1}{2}\sigma_{k+1}^*\sigma'^*_{k+1}\right)\Delta_+\right),$$

$$G_m(X_{t_{k+1}}^*) = \sigma_{k+1}^*(\sigma'^*_{k+1})^2\Delta_+,$$

for $E_m(X_{t_{k+1}}^*) \geq 0$ (which cannot be rearranged for $X_{t_{k+1}}^*$ in general); otherwise, the density is equal to zero. The terms $\mu_{k+1}^*$ and $\sigma_{k+1}^*$ are similar to $\mu_{k+1}$ and $\sigma_{k+1}$, but $X_{t_{k+1}}$ is replaced by $X_{t_{k+1}}^*$. Here, we do not respectively approximate $\mu_{k+1}$ and $\sigma_{k+1}$ by $\mu_k$ and $\sigma_k$ because doing so does not lead to simplification. Moreover, there is no closed formula for the normalization constant needed to scale the product of the two transition densities to a proper density.

For the GBM, we have $X_t > 0$ and $\sigma_{k+1}^* = \sigma X_{t_{k+1}}^* > 0$ and thus, obtain the following bounds for $\pi^{\mathrm{Mil}}(X_{t_m}|X_{t_{k+1}}^*, \theta)$, the second factor in (3.4):

$$X_{t_{k+1}}^* \leq \frac{X_{t_m}}{\frac{1}{2} + (\alpha - \frac{1}{2}\sigma^2)\Delta_+} =: u_{\mathrm{2nd}}, \quad \text{if } \frac{1}{2} + \left(\alpha - \frac{1}{2}\sigma^2\right)\Delta_+ > 0 \quad \text{(Case I)},$$

$$X_{t_{k+1}}^* \geq \frac{X_{t_m}}{\frac{1}{2} + (\alpha - \frac{1}{2}\sigma^2)\Delta_+} =: l_{\mathrm{2nd}}, \quad \text{if } \frac{1}{2} + \left(\alpha - \frac{1}{2}\sigma^2\right)\Delta_+ < 0 \quad \text{(Case II)}$$

and

$$X_{t_{k+1}}^* \geq 0, \quad \text{if } \frac{1}{2} + \left(\alpha - \frac{1}{2}\sigma^2\right)\Delta_+ = 0 \quad \text{(Case III)}.$$

From (2.12), we obtain the following lower bound for $\pi^{\mathrm{Mil}}(X_{t_{k+1}}^* | X_{t_k}^*, \theta)$, the first factor in (3.4):

$$X_{t_{k+1}}^* \geq X_{t_k}^*\left(\frac{1}{2} + \left(\alpha - \frac{1}{2}\sigma^2\right)\Delta t_k\right) =: l_{\mathrm{1st}}.$$

At the same time, proposals $X_{t_{k+1}}^*$ for the GBM should always be strictly positive to be in the state space. Let $l := \max\{0, l_{\mathrm{1st}}\}$. The constraints on $X_{t_{k+1}}^*$ derived from the two factors in (3.4) lead to three cases for the set $\mathcal{D}$ of feasible points of $X_{t_{k+1}}^*$ for the GBM (assuming $X_{t_m} > 0$)

$$\mathcal{D} = \begin{cases} \emptyset, & \text{if (Case I) applies and } l_{\mathrm{1st}} > u_{\mathrm{2nd}}, \\ [l, u_{\mathrm{2nd}}], & \text{if (Case I) applies and } l_{\mathrm{1st}} \leq u_{\mathrm{2nd}}, \\ [l, \infty), & \text{if (Case II) or (Case III) apply.} \end{cases}$$

Since the MB proposal takes into account information not only from the left data point but also from the observation on the right, it does not have a large jump in the last step as the left-conditioned proposal does. This is also apparent in the simulations in figure 4e,f. Therefore, the acceptance probability and acceptance rate are usually higher for the MB proposal than for the left-conditioned proposal. As appendix B demonstrates, the acceptance probability is even equal to 1 for the MB proposal if only one data point is imputed between two observations (i.e. the number of inter-observation intervals is $m = 2$). This holds when using the Milstein scheme to approximate the transition density for the likelihood function and proposal density, but also when using the Euler scheme without the approximation of $\mu_{k+1}$ and $\sigma_{k+1}$ by $\mu_k$ and $\sigma_k$, respectively.

The density of the MB proposal based on the Euler scheme in equation (3.5) can also be interpreted as the density that results from applying the Euler scheme to the following diffusion process:

$$dX_t = \left(\frac{X_{\tau_{i+1}} - X_t}{\tau_{i+1} - t}\right)dt + \sqrt{\frac{\tau_{i+1} - t_{k+1}}{\tau_{i+1} - t}}\sigma(X_t, \theta)\,dB_t$$

for $t \in [t_k, t_{k+1}]$. See [10] for a detailed discussion of the connection between the modified bridge and the continuous-time conditioned process. Applying the Milstein scheme to this process yields another proposal scheme to which we refer as the diffusion bridge Milstein (DBM) proposal. For the DBM proposal, the proposal density of a path segment also factorizes as:

$$q_{\mathrm{DBM}}(X_{(\tau_i, \tau_{i+1})}^{\mathrm{imp}*} \,|\, X_{\tau_i}, X_{\tau_{i+1}}, \theta) = \prod_{k=0}^{m-2} \pi(X_{t_{k+1}}^* \,|\, X_{t_k}^*, X_{\tau_{i+1}}, \theta),$$

where $X_{t_0}^* = X_{\tau_i}$, and each factor $\pi(X_{t_{k+1}}^* | X_{t_k}^*, X_{\tau_{i+1}}, \theta)$ corresponds to the density based on the Milstein scheme from §2.2 where we replace $\mu_k$ by $(X_{\tau_{i+1}} - X_{t_k})/(\tau_{i+1} - t_k)$, $\sigma_k$ by

$\sqrt{(\tau_{i+1} - t_{k+1})/(\tau_{i+1} - t_k)}\sigma(X_{t_k}, \theta)$, and $\sigma_k'$ by $\sqrt{(\tau_{i+1} - t_{k+1})/(\tau_{i+1} - t_k)}\partial\sigma(y, \theta)/\partial y|_{y=X_{t_k}}$. Like the MB proposal, the DBM proposal takes into account information from the observation on the right, and, therefore, it does not have a large jump in the last step as illustrated in figure 4g.

Thus far, our path update has only been applied to imputed points between two observations. It can easily be extended to a case with several observations along the path by simply decomposing the path into independent path proposals, multiplying the respective acceptance probabilities and collectively accepting or rejecting the proposals. Moreover, the entire path does not have to be updated all at once, but can be divided into several path segments that are successively updated. Different algorithms for choosing the update interval are summarized in [2] and appendix C describes one of them.

Another challenge in the context of Bayesian data augmentation and the MCMC scheme discussed above is the dependence between the parameter components included in the diffusion function and the missing path segments between two observations. Roberts & Stramer [5] were the first to highlight that, in the discretized setting (as we consider it here), the dependence leads to a slower convergence of the MCMC algorithm as the number of imputed points $m$ increases. All estimation methods compared here are affected by this issue in the same way; we hence do not further consider it here.

We have introduced a number of possible options for the choices to be made when constructing an estimation method in the framework as described so far:

— approximate the transition densities in the likelihood function based on the Euler or Milstein scheme,
— use the left-conditioned, the MB or the DBM proposal, and
— use the Euler or Milstein scheme for the proposal densities (for the left-conditioned or MB proposal).

In the following, we will omit the left-conditioned proposal due to the inefficiency that we already pointed out. Instead, we will consider the following four combinations:

(**MBE-E**)   MB proposal and transition density both based on the Euler scheme,
(**MBE-M**)   MB proposal based on the Euler scheme and transition density based on the Milstein scheme,
(**MBM-M**)  MB proposal and transition density both based on the Milstein scheme, and
(**DBM-M**)   DBM proposal (which is based on the Milstein scheme) and transition density based on the Milstein scheme.

Combination MBE-M merges the Euler and Milstein scheme. We include it here because it combines the faster scheme for the proposals (where accuracy is less important) and the more accurate scheme for the acceptance probability.

To our knowledge, we are the first to use the Milstein scheme in the MCMC context described above.

## 4. Implementation

The implementation is relatively straightforward for the majority of the estimation procedures, and only the combination of the MB proposal and the Milstein approximation requires additional explanation. As mentioned, when approximating the two factors on the right-hand side of (3.4) by the transition density based on the Milstein scheme, there is no closed formula for the normalization constant to obtain a proper density. The normalization is necessary because the proposal density for a path segment is the product of several of the terms from (3.4), where the condition on the left point, $X_{t_k}^*$, differs between a newly proposed segment and the last accepted segment if several consecutive points are imputed. Therefore, the normalization constants differ and do not cancel out in the acceptance probability. Normalization is not necessary only in the case where just one point is imputed between two observations (i.e. $m = 2$ subintervals) because the left point, $X_{t_k}$, is always a (fixed) observed point that is not updated. Thus, the normalization constants cancel out in the acceptance probability. For $m > 2$, we numerically integrate the product (3.4) over $X_{t_{k+1}}$ to obtain the normalization constant. The product in (3.4) may be very small (but not zero everywhere in a non-empty feasible set $\mathcal{D}$) and may thus numerically integrate to zero, especially when the upper interval bound of the feasible set is infinite. To overcome this problem, we take two measures. First, we do not integrate over the entire set of feasible points but determine the maximum of the product numerically and then integrate over the interval that includes all points with a function value of at least $10^{-20}$ times this maximum. Second, we rescale the product in (3.4) by dividing by the maximum before integrating.

To sample from the Milstein MB proposal density, we employ rejection sampling. For this, normalization of the product in (3.4) is not necessary. Again, we numerically determine the maximum $d_{\max}$ of the product, and the interval $\mathcal{I}$ that includes all points with a function value of at least $10^{-20}$ times this maximum. Then, we uniformly sample $(u_1, u_2)$ from rectangle $\mathcal{I} \times (0, d_{\max})$ and accept $u_1$ as a proposal $X^*_{t_{k+1}}$ if the unnormalized density value of (3.4) at $u_1$ is at most $u_2$.

For the combination of the MB proposal and the Milstein approximation, the set of feasible proposal points may be empty. In this case, our implementation shifts to the Euler approximation for this point, i.e. the point is proposed with the MB proposal based on the Euler scheme and also the corresponding factor of the proposal density in the acceptance probability is based on the Euler scheme. In addition, for all methods, a negative point may be proposed, which is not feasible for a GBM. Therefore, in this case, we propose a new point. For both cases, we count the number of times that they occur during the estimation procedure. In the following simulation study, no cases of switching to the Euler scheme occurred and negative proposals occurred only very rarely (less than 1‰ of the number of iterations in the very worst case).

We implemented the described estimation procedures in R v. 3.6.2 [23]. The source code of our implementation and the following simulation study is publicly available at https://github.com/fuchslab/Inference_for_SDEs_with_the_Milstein_scheme.

# 5. Simulation study

In this section, we study the computational performance of the competing inference methods on the (relatively simple) benchmark model GBM. As a second (on the application side more often studied) benchmark model, the CIR process is investigated in appendix D. In this work, we focus on Bayesian inference by data augmentation and compare the four approaches listed at the end of §3.2. Conceptually different inference procedures, as summarized e.g. in [2], are not considered as competitors here as they would be employed in different data contexts. There are two aspects that are important to consider when we want to evaluate the different methods:

(a) the accuracy with which the true posterior distribution is approximated based on one of the approximation schemes and a given number $m$ and
(b) the accuracy with which we are able to draw from this approximated posterior distribution.

We are interested in the overall accuracy, i.e. the combination of (a) and (b), achieved within a fixed amount of computational time.

For the simulation study, we generated 100 paths of the GBM in the time interval $[0, 1]$ using the solution (2.3) with the parameter combination $\theta = (\alpha, \sigma^2)^{\mathrm{T}} = (1, 2)^{\mathrm{T}}$ and initial value $x_0 = 100$. Figure 5 illustrates some of these paths. From each path, we took $M = 20$ equidistant points (i.e. the inter-observation time $\Delta t$ was 0.05) and applied each of the four described estimation methods once. We imputed data such that we got $m = 2$ and $m = 5$ inter-observation intervals. We also included the case $m = 1$, i.e. no data were imputed and only Step (1) from §3, the parameter update, was repeated in the estimation procedure where the likelihood of the path in the acceptance probability is approximated by the Euler or the Milstein scheme. For the prior distribution of the parameters, we assumed that they were independently distributed with $\alpha \sim \mathcal{N}(0, 10)$ and $\sigma^2 \sim \mathrm{IG}(\kappa_0 = 2, \nu_0 = 2)$, where IG denotes the inverse gamma distribution with shape parameter $\kappa_0$ and scale parameter $\nu_0$. The *a priori* expectations of the parameters are thus $\mathbb{E}(\alpha) = 0$ and $\mathbb{E}(\sigma^2) = 2$.

Each of the estimation procedures performs the following steps:

(1) Draw initial values for the parameters $\alpha$ and $\sigma^2$ from the prior distributions.
(2) Initialize $Y^{\mathrm{imp}}$ by linear interpolation.
(3) Repeat the following steps:

 — Parameter update: Apply random walk proposals.
 (a) Draw a proposal $\alpha^* \sim \mathcal{N}(\alpha_{i-1}, 0.25)$.
 (b) Draw a proposal $\sigma^{2*} \sim \mathcal{LN}(\log \sigma^2_{i-1}, 0.25)$.
 (c) Accept both or none.
 — Path update:
 (a) Choose an update interval $(t_a, t_b)$ as described in appendix C with $\lambda = 5$.
 (b) Draw a proposal $X^{\mathrm{imp}*}_{(t_a, t_b)}$ according to the investigated method.
 (c) Accept or reject the proposal.

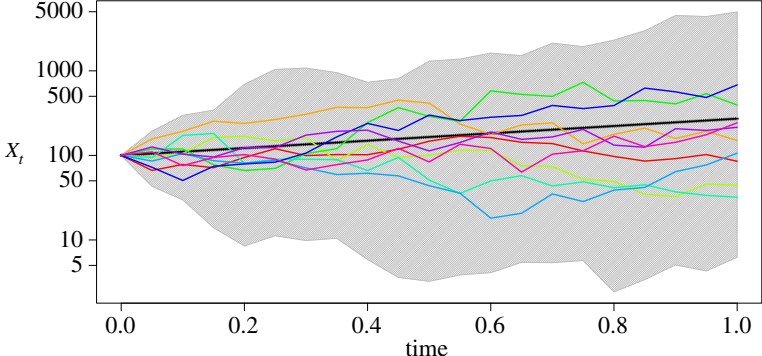

**Figure 5.** Trajectories used in the simulation study. The solid black line represents the expected value of the GBM solution $\mathbb{E}[X_t] = X_0 \exp(\alpha t) = 100 \exp(t)$. The coloured lines are 10 examples of the 100 trajectories used in the simulation study. The grey-shaded area shows the range of the 100 trajectories. Each trajectory consists of 20 points used as observations.

We let each procedure run for 1 h and evaluate the overall accuracy of the obtained sample compared to a sample from the true posterior distribution (as described below).

Figures 6 and 7 present the output from one estimation procedure on the example of the combination MBM-M of the MB proposal and the Milstein approximation for the proposal density and the likelihood function. From each estimation procedure, we obtained an MCMC chain of dimension $n(m-1)+2$. For each chain, we used the two components for parameters $\alpha$ and $\sigma^2$ and calculated the mean, the median, and the variance after cutting off a burn-in phase of 5000 iterations. To justify our use of independent proposals for the parameter update, we show in appendix E that the parameters are not strongly correlated.

As a benchmark, we also sampled from the true parameter posterior distribution based on the solution of the GBM. We used the Stan software [24,25] which provides an efficient C++ implementation of Hamiltonian Monte Carlo (HMC) sampling with the No-U-turn sampler to sample from the true parameter posterior distribution. For each posterior distribution corresponding to one of the 100 sample paths, we generated four HMC chains with 500 000 iterations each. The first half of the chains was discarded as warm-up and the remaining draws were combined to give a sample of size $10^6$. We calculated the multivariate effective sample size (ESS) as defined in [26] which provides the size of an independent and identically distributed sample equivalent to our samples in terms of variance and found that the ESS of the obtained samples from the true posterior distribution is well over 500 000. For each of these samples, we also calculated the mean, the median and the variance.

The estimation procedures and time measurements were performed on a cluster of machines with the following specifications: AMD Opteron™ Processor 6376 (1.40 GHz), 512GB DDR3-RAM.

# 6. Results

Figures 8 and 9 and tables 1 and 2 summarize the results of running each of the methods once for 1 h for each of the 100 GBM trajectories. Figures 8 and 9 show the density plots of the difference between the respective statistic (mean, median or variance) calculated for a sample from the approximated posterior distribution obtained by the respective method and the statistic for a sample from the true posterior distribution of the same sample path. Each density plot aggregates 100 such difference values, one for each of the 100 GBM trajectories. Table 1 tabulates the root mean square error (RMSE) based on these differences for each of the considered methods, discretization levels $m$ and statistics. We use the RMSE as the measure of the overall accuracy. The lower the RMSE is, the higher the accuracy of the respective method. Table 2 empirically evaluates the computational efficiency of the considered methods, including the number of iterations completed after 1 h, the multivariate ESS based on the obtained sample after discarding a burn-in phase of 5000 iterations, and the acceptance rates of the parameter and the path proposals. Each of these quantities is averaged over the 100 GBM trajectories and the coefficient of variation is also stated.

For the drift parameter $\alpha$ of the GBM, the four considered schemes perform comparably for $m=2$ and $m=5$. In particular, the use of the Milstein approximation does not improve the accuracy of the posterior

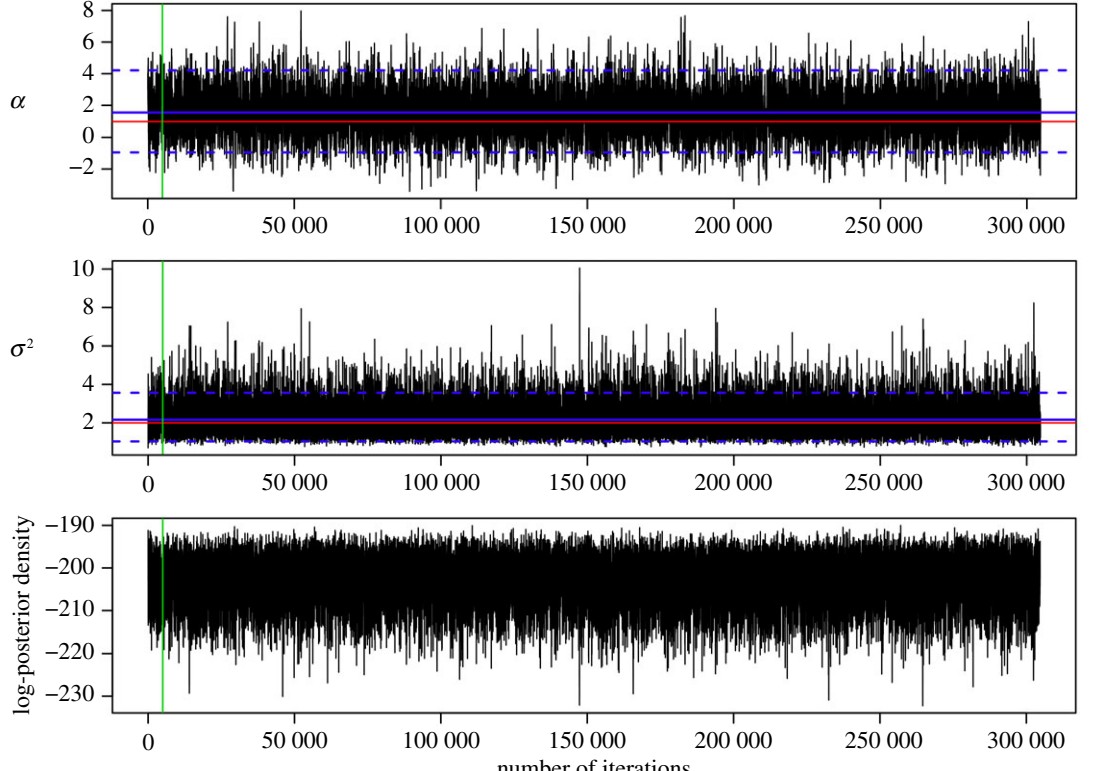

**Figure 6.** Trace plots of the MCMC chains for parameters $\alpha$ and $\sigma^2$ of the GBM (2.2) and of the log-posterior density values for one parameter estimation run using the combination MBM-M of the modified bridge proposal with $m = 2$ and the Milstein approximation for the proposal density and the likelihood function. The red lines represent the true values of parameters $\alpha = 1$ and $\sigma^2 = 2$, the blue solid lines represent the mean, and the blue dashed lines represent the lower and upper bounds of the highest-probability density interval of 95% after cutting off the first 5000 values of the chains as burn-in, which is represented by the green line.

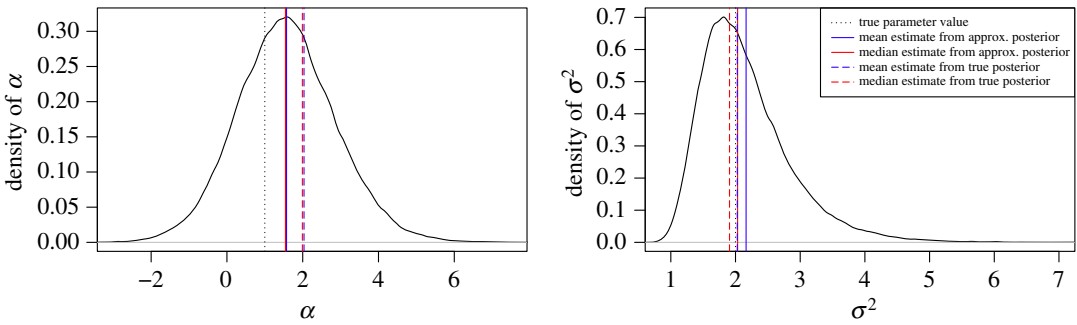

**Figure 7.** Estimated posterior densities for $\alpha$ and $\sigma^2$ from one parameter estimation run using the combination MBM-M of the modified bridge proposal and the Milstein approximation for the proposal and the transition density. Moreover, true values of the parameters, the mean and the median of the MCMC chains after 5000 iterations burn-in, and the mean and the median of a sample from the true posterior distribution of the sample path based on the solution of the GBM are shown.

mean and median for the same discretization level $m$. The accuracy of the posterior variance is slightly improved by the use of the Milstein approximation when data are imputed. Moreover, for MBE-E, the accuracy does not consistently improve as $m$ is increased. Whereas, the accuracy for the methods including the Milstein scheme improves considerably when imputed data are introduced (i.e. $m > 1$) and it improves slightly when $m$ is increased from 2 to 5.

For the diffusion parameter $\sigma^2$ of the GBM, we clearly see an improvement in overall accuracy for the methods involving the Milstein scheme. Combination DBM-M turns out to be the most accurate, closely followed by MBE-M in case of the mean and median.

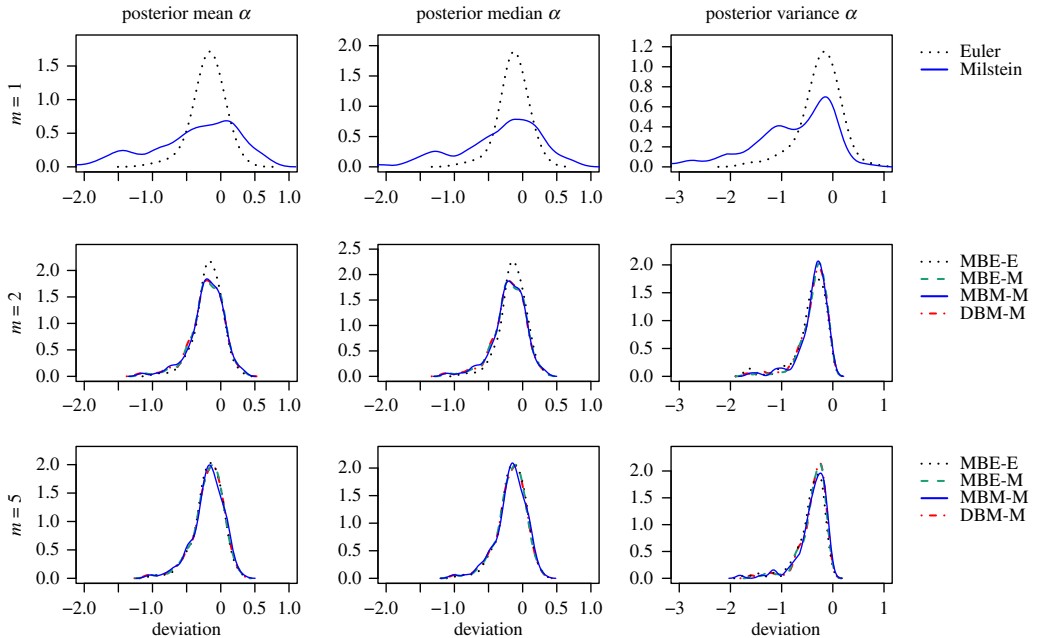

**Figure 8.** Sampling results for $\alpha$ obtained by each of the estimation procedures. Each density plot aggregates 100 deviations between the respective statistics (left: mean, middle: median, right: variance) calculated for the sample from the approximated posterior and for the sample from the true posterior distribution, one for each of the 100 sample paths of the GBM. The rows show results for different numbers $m$ of subintervals between two observations. For $m = 1$, no data points were imputed and only Step (1), the parameter update, was repeated in the estimation procedure.

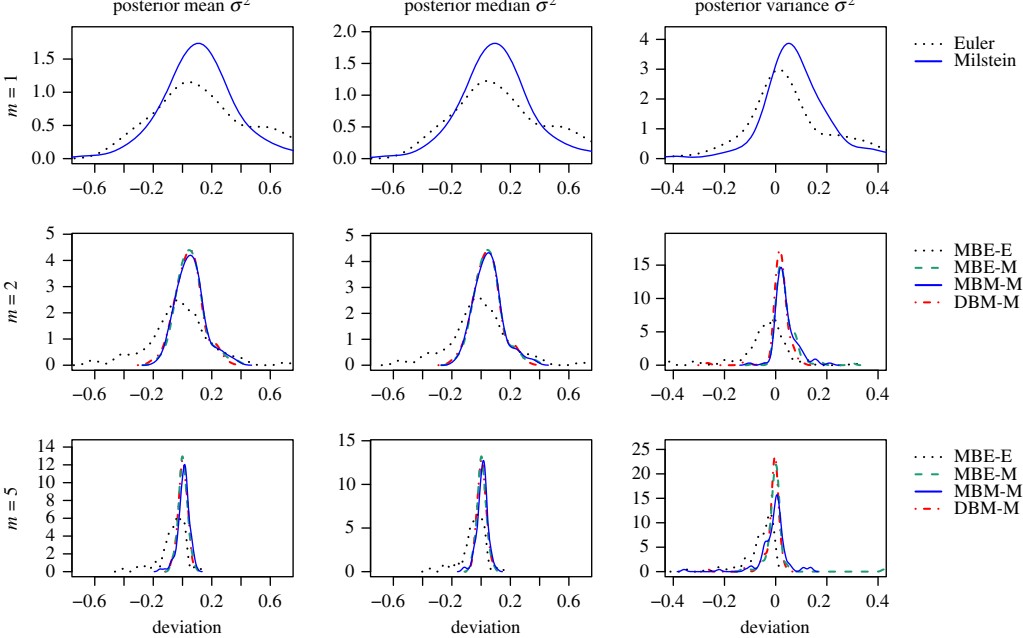

**Figure 9.** Sampling results for $\sigma^2$ as described in figure 8.

According to table 2, the number of iterations completed within 1 h varies substantially among the different estimation procedures. It is always higher for the procedures that use the Euler approximation, while especially combination MBM-M is very time-consuming and thus completes fewer iterations. Similarly, the multivariate ESS varies substantially among the different estimation

**Table 1.** Empirical characteristics for evaluating the overall accuracy of the parameter estimation procedures for different numbers $m$ of subintervals between two observations aggregated over 100 deviations between the respective statistics calculated for the sample from the approximated posterior and for the sample from the true posterior distribution, one for each of the 100 sample paths of the GBM. The lowest RMSE (root mean square error) per $m$ and per statistic is printed in bold.

|  | method | RMSEs for $\alpha$ | | | RMSEs for $\sigma^2$ | | |
|---|---|---|---|---|---|---|---|
|  |  | mean | median | variance | mean | median | variance |
| $m = 1$ | Euler | **0.282** | **0.244** | **0.456** | 0.638 | 0.600 | 0.471 |
|  | Milstein | 0.851 | 0.780 | 1.158 | **0.282** | **0.265** | **0.176** |
| $m = 2$ | MBE-E | **0.266** | **0.238** | 0.526 | 0.211 | 0.198 | 0.141 |
|  | MBE-M | 0.311 | 0.302 | 0.476 | 0.109 | 0.106 | 0.057 |
|  | MBM-M | 0.315 | 0.305 | **0.470** | 0.112 | 0.107 | 0.057 |
|  | DBM-M | 0.318 | 0.308 | 0.485 | **0.101** | **0.099** | **0.044** |
| $m = 5$ | MBE-E | **0.277** | **0.254** | 0.524 | 0.113 | 0.098 | 0.127 |
|  | MBE-M | 0.288 | 0.274 | 0.474 | 0.031 | 0.031 | 0.050 |
|  | MBM-M | 0.292 | 0.278 | 0.492 | 0.040 | 0.037 | 0.058 |
|  | DBM-M | 0.291 | 0.275 | **0.472** | **0.031** | **0.030** | **0.037** |

**Table 2.** Empirical characteristics for evaluating the computational efficiency of the parameter estimation procedures for different numbers $m$ of subintervals between two observations aggregated over 100 trajectories of the GBM. Each of the procedures was run for 1 h. Acceptance rates are defined to take values between 0 and 1. For $m = 1$, no data points were imputed and only Step (1), the parameter update, was repeated in the estimation procedure. Specifications for the computing power are stated in the main text. c.v. denotes the coefficient of variation.

|  | method | number of iterations after 1 h | | multivariate effective sample size | | acceptance rate of the parameters | | acceptance rate of the path | |
|---|---|---|---|---|---|---|---|---|---|
|  |  | mean | c.v. | mean | c.v. | mean | c.v. | mean | c.v. |
| $m = 1$ | Euler | 25 134 301 | 0.03 | 1 273 744 | 0.16 | 0.518 | 0.02 | — | — |
|  | Milstein | 454 863 | 0.03 | 146 362 | 0.41 | 0.425 | 0.14 | — | — |
| $m = 2$ | MBE-E | 8 583 614 | 0.03 | 170 827 | 0.19 | 0.442 | 0.01 | 0.842 | 0.04 |
|  | MBE-M | 1 816 144 | 0.03 | 24090 | 0.38 | 0.417 | 0.03 | 0.799 | 0.05 |
|  | MBM-M | 300 870 | 0.03 | 6881 | 0.21 | 0.417 | 0.03 | 1.000 | 0.00 |
|  | DBM-M | 754 024 | 0.10 | 28 089 | 0.31 | 0.417 | 0.03 | 0.839 | 0.04 |
| $m = 5$ | MBE-E | 6 765 054 | 0.10 | 49 885 | 0.18 | 0.310 | 0.01 | 0.892 | 0.02 |
|  | MBE-M | 892 487 | 0.02 | 5033 | 0.24 | 0.304 | 0.01 | 0.844 | 0.03 |
|  | MBM-M | 78 215 | 0.04 | 573 | 0.20 | 0.304 | 0.01 | 0.978 | 0.01 |
|  | DBM-M | 879 227 | 0.03 | 5535 | 0.21 | 0.304 | 0.01 | 0.884 | 0.02 |

procedures. It is higher for $m = 2$ than for $m = 5$ for each of the considered estimation procedures. The acceptance rate of the parameters is slightly lower when the Milstein scheme is used for the approximation of the likelihood function. In addition, the acceptance rate of the parameters decreases as the number of imputed points increases. The acceptance rate of the path is highest for combination MBM-M. For MBE-E, it would be just as high if one would not substitute $\mu_{k+1}$ and $\sigma_{k+1}$ by $\mu_k$ and $\sigma_k$. For MBE-E, MBE-M and DBM-M, the acceptance rate of the path increases as the number of imputed points increases.

# 7. Summary and discussion

We have demonstrated how to implement an algorithm for the parameter estimation of SDEs from low-frequency data using the Milstein scheme to approximate the transition density of the underlying process. Our motivation was to improve numerical accuracy and thus reduce the amount of imputed data and computational overhead. However, our findings are rather discouraging: we found that this method can be applied to multidimensional processes only with impractical restrictions. Moreover, we showed that the combination of the MB proposal with the Milstein scheme for the proposal density may lead to an empty set of possible proposal points, which would require switching to the Euler scheme in order to proceed. One of the strengths of the original (Euler-based) MCMC scheme is its generic character and applicability. Through this, it possesses a practical advantage over otherwise more sophisticated methods such as the Exact Algorithm [27]. This strength does not translate to the Milstein-based MCMC scheme due to the limited applicability of the Milstein approximation especially in the multidimensional setting. Thus, methods like the Exact Algorithm may be a reasonable alternative. The limited applicability of the Milstein approximation would also persist for advanced forms of the discussed MCMC scheme like the innovation scheme in [6] or for even more generic algorithms like particle MCMC as studied in [28].

In our simulation study, we found that the overall accuracy for the estimates for the drift parameter of the GBM does not necessarily improve when the Milstein scheme is used. Fewer iterations are completed for the methods involving the Milstein scheme and also the ESS is substantially lower. Thus, the poor sampling efficiency might outweigh the (potential) increase in accuracy of the approximation of the posterior distribution. Especially, the combination MBM-M results in a particularly low number of iterations and a low ESS. Owing to the already quite low ESS achieved by the Milstein-based methods for $m = 5$ subintervals between two observations, we did not consider higher discretization levels. Moreover, note that tuning the variance hyperparameters for the random walk proposals of the parameters in Step (3) in the simulation study to reach an optimal acceptance rate might lead to a higher ESS. However, since the acceptance rates achieved in the simulation study lie in a range where the sampling efficiency is rather robust to changes in the acceptance rate as shown in [29] (in the high-dimensional limit), we do not expect the change in the ESS after tuning to be substantial.

For the estimates for the GBM diffusion parameter, the overall accuracy is increased by the use of the Milstein scheme. DBM-M turns out to be the most effective combination in terms of overall accuracy.

We conducted another simulation study on the example of the CIR process, as shown in appendix D, and the results are very similar as for the GBM. The use of the Milstein approximation does not consistently improve the overall accuracy for the drift parameter; however, it does improve the accuracy for the diffusion parameter. Again combination DBM-M achieves the highest accuracy, closely followed by MBE-M.

It was expected that the use of the Milstein scheme would make a difference for the estimates for the diffusion parameters because the additional term added by the Milstein scheme compared to the Euler scheme involves the diffusion function and its derivative. Nevertheless, the general applicability of the Euler scheme remains a great advantage and the search for different proposal schemes such as in [10,22] rather than for different numerical discretization schemes may be a more promising way towards more efficient estimation algorithms for diffusion processes.

Data accessibility. The source code of our implementation and the simulation study is publicly available at https://github.com/fuchslab/Inference_for_SDEs_with_the_Milstein_scheme.

Authors' contributions. C.F. devised the project and provided supervision. S.P. implemented the described algorithms, carried out the simulation study and drafted the manuscript. Both authors contributed to the final version of the manuscript, gave final approval for publication and agree to be held accountable for the work performed therein.

Competing interests. The authors declare that there are no conflicts of interest regarding the publication of this paper.

Funding. Our research was supported by the German Research Foundation within the SFB 1243, Subproject A17, by the Federal Ministry of Education and Research under grant no. 01DH17024, and by the Helmholtz pilot project 'Uncertainty Quantification'.

Acknowledgements. The authors wish to thank three anonymous reviewers for very valuable suggestions that helped to significantly improve this article.

# Appendix A. Derivation of the transition density based on the Milstein scheme

The Milstein scheme

$$Y_{k+1} = Y_k + \mu(Y_k, \theta)\Delta t_k + \sigma(Y_k, \theta)\Delta B_k + \frac{1}{2}\sigma(Y_k, \theta)\frac{\partial \sigma}{\partial y}(Y_k, \theta)((\Delta B_k)^2 - \Delta t_k),$$

can be considered a variable transformation of the random variable $Z \sim \mathcal{N}(0, 1)$ with density $\phi(z)$ using the transformation function

$$f(z) = az^2 + bz + c,$$

where the coefficients are defined as

$$a = \frac{1}{2}\sigma(Y_k, \theta)\frac{\partial \sigma}{\partial y}(Y_k, \theta)\Delta t_k,$$

$$b = \sigma(Y_k, \theta)\sqrt{\Delta t_k}$$

and
$$c = Y_k + \left[\mu(Y_k, \theta) - \frac{1}{2}\sigma(Y_k, \theta)\frac{\partial \sigma}{\partial y}(Y_k, \theta)\right]\Delta t_k,$$

and whose derivative and inverse function are

$$f'(z) = 2az + b$$

and

$$f^{-1}(y) = -\frac{b}{2a} \pm \frac{\sqrt{b^2 + 4a(y - c)}}{2a} \quad \text{for } y \geq -\frac{b^2}{4a} + c.$$

By applying the random variable transformation theorem as found in [30, p. 269] or [31, p. 27], the density $\rho_Y$ of $Y_{k+1}$ can be derived as follows:

$$\rho_Y(y) = \sum_{\{z \in \mathbb{R}: f(z) = y\}} \frac{\phi(z)}{|f'(z)|}$$

$$= \frac{\phi\left(-\frac{b}{2a} - \frac{\sqrt{b^2 + 4a(y-c)}}{2a}\right)}{\left|f'\left(-\frac{b}{2a} - \frac{\sqrt{b^2 + 4a(y-c)}}{2a}\right)\right|} + \frac{\phi\left(-\frac{b}{2a} + \frac{\sqrt{b^2 + 4a(y-c)}}{2a}\right)}{\left|f'\left(-\frac{b}{2a} + \frac{\sqrt{b^2 + 4a(y-c)}}{2a}\right)\right|}$$

$$= \frac{\frac{1}{\sqrt{2\pi}}\exp\left(-\frac{1}{2}\left(-\frac{b}{2a} - \frac{\sqrt{b^2 + 4a(y-c)}}{2a}\right)^2\right)}{\left|b + 2a\left(-\frac{b}{a} - \frac{\sqrt{b^2 + 4a(y-c)}}{2a}\right)\right|} + \frac{\frac{1}{\sqrt{2\pi}}\exp\left(-\frac{1}{2}\left(-\frac{b}{2a} + \frac{\sqrt{b^2 + 4a(y-c)}}{2a}\right)^2\right)}{\left|b + 2a\left(-\frac{b}{2a} + \frac{\sqrt{b^2 + 4a(y-c)}}{2a}\right)\right|}$$

$$= \frac{1}{\sqrt{2\pi}}\left(\frac{\exp\left(-\frac{1}{8a^2}\left(b^2 + 2b\sqrt{b^2 + 4a(y-c)} + b^2 + 4a(y-c)\right)\right)}{\left|-\sqrt{b^2 + 4a(y-c)}\right|}\right.$$
$$\left. + \frac{\exp\left(-\frac{1}{8a^2}\left(b^2 - 2b\sqrt{b^2 + 4a(y-c)} + b^2 + 4a(y-c)\right)\right)}{\left|\sqrt{b^2 + 4a(y-c)}\right|}\right)$$

$$= \frac{\exp\left(-\frac{b^2 + 2a(y-c)}{4a^2}\right)}{\sqrt{2\pi}\sqrt{b^2 + 4a(y-c)}}\left(\exp\left(-\frac{b\sqrt{b^2 + 4a(y-c)}}{4a^2}\right) + \exp\left(\frac{b\sqrt{b^2 + 4a(y-c)}}{4a^2}\right)\right)$$

$$= \frac{\exp\left(-\frac{b^2 + 2a(y-c)}{4a^2}\right)}{\sqrt{2\pi}\sqrt{b^2 + 4a(y-c)}} \cdot 2\cosh\left(\frac{b\sqrt{b^2 + 4a(y-c)}}{4a^2}\right).$$

After substituting the coefficients $a$, $b$ and $c$ and abbreviating $\mu_k := \mu(Y_k, \theta)$, $\sigma_k := \sigma(Y_k, \theta)$ and

$\sigma'_k := \sigma'(Y_k, \theta) = \partial \sigma(y, \theta)/\partial y\big|_{y=Y_k}$, we obtain the transition density based on the Milstein scheme

$$\pi^{Mil}(Y_{k+1}|Y_k, \theta) = \frac{\exp\left(-\frac{\left(\sigma_k\sqrt{\Delta t_k}\right)^2 + 2\frac{1}{2}\sigma_k\sigma'_k\Delta t_k(Y_{k+1} - Y_k - (\mu_k - \frac{1}{2}\sigma_k\sigma'_k)\Delta t_k)}{4(\frac{1}{2}\sigma_k\sigma'_k\Delta t_k)^2}\right)}{\sqrt{2\pi}\sqrt{\left(\sigma_k\sqrt{\Delta t_k}\right)^2 + 4\frac{1}{2}\sigma_k\sigma'_k\Delta t_k(Y_{k+1} - Y_k - (\mu_k - \frac{1}{2}\sigma_k\sigma'_k)\Delta t_k)}}$$

$$\cdot \left[\exp\left(-\frac{\sigma_k\sqrt{\Delta t_k}\sqrt{\left(\sigma_k\sqrt{\Delta t_k}\right)^2 + 4\frac{1}{2}\sigma_k\sigma'_k\Delta t_k(Y_{k+1} - Y_k - (\mu_k - \frac{1}{2}\sigma_k\sigma'_k)\Delta t_k)}}{4(\frac{1}{2}\sigma_k\sigma'_k\Delta t_k)^2}\right)\right.$$

$$\left. + \exp\left(\frac{\sigma_k\sqrt{\Delta t_k}\sqrt{\left(\sigma_k\sqrt{\Delta t_k}\right)^2 + 4\frac{1}{2}\sigma_k\sigma'_k\Delta t_k(Y_{k+1} - Y_k - (\mu_k - \frac{1}{2}\sigma_k\sigma'_k)\Delta t_k)}}{4(\frac{1}{2}\sigma_k\sigma'_k\Delta t_k)^2}\right)\right]$$

$$= \frac{\exp\left(-\frac{C_k(Y_{k+1})}{D_k}\right)}{\sqrt{2\pi}\sqrt{\Delta t_k}\sqrt{A_k(Y_{k+1})}} \cdot \left[\exp\left(-\frac{\sqrt{A_k(Y_{k+1})}}{D_k}\right) + \exp\left(\frac{\sqrt{A_k(Y_{k+1})}}{D_k}\right)\right]$$

with

$$A_k(Y_{k+1}) = (\sigma_k)^2 + 2\sigma_k\sigma'_k\left(Y_{k+1} - Y_k - \left(\mu_k - \frac{1}{2}\sigma_k\sigma'_k\right)\Delta t_k\right)$$

$$C_k(Y_{k+1}) = \sigma_k + \sigma'_k\left(Y_{k+1} - Y_k - \left(\mu_k - \frac{1}{2}\sigma_k\sigma'_k\right)\Delta t_k\right)$$

$$D_k = \sigma_k(\sigma'_k)^2\Delta t_k$$

and for

$$Y_{k+1} \geq Y_k - \frac{1}{2}\frac{\sigma_k}{\sigma'_k} + \left(\mu_k - \frac{1}{2}\sigma_k\sigma'_k\right)\Delta t_k, \quad \text{if } \sigma_k\sigma'_k > 0$$

and

$$Y_{k+1} \leq Y_k - \frac{1}{2}\frac{\sigma_k}{\sigma'_k} + \left(\mu_k - \frac{1}{2}\sigma_k\sigma'_k\right)\Delta t_k, \quad \text{if } \sigma_k\sigma'_k < 0.$$

In the case of $\sigma_k = 0$, $Y_{k+1}$ conditioned on $Y_k$ is deterministic. For $\sigma'_k = 0$, the Milstein scheme reduces to the Euler scheme.

# Appendix B. Derivation of the acceptance probability for the MB proposal for $m = 2$ inter-observation intervals

As stated in §3.2, the acceptance probability for the path update between two consecutive observations $X_{\tau_i}$ and $X_{\tau_{i+1}}$ with the MB proposal is

$$\zeta(X^{imp*}_{(\tau_i,\tau_{i+1})}, X^{imp}_{(\tau_i,\tau_{i+1})}) = 1 \wedge \frac{\pi(X^{imp*}_{(\tau_i,\tau_{i+1})} \mid X^{obs}_{\{\tau_i,\tau_{i+1}\}}, \theta)q_{MB}(X^{imp}_{(\tau_i,\tau_{i+1})} \mid X_{\tau_i}, X_{\tau_{i+1}}, \theta)}{\pi(X^{imp}_{(\tau_i,\tau_{i+1})} \mid X^{obs}_{\{\tau_i,\tau_{i+1}\}}, \theta)q_{MB}(X^{imp*}_{(\tau_i,\tau_{i+1})} \mid X_{\tau_i}, X_{\tau_{i+1}}, \theta)}$$

$$= 1 \wedge \prod_{k=0}^{m-1}\frac{\pi(X^*_{t_{k+1}} \mid X^*_{t_k}, \theta)}{\pi(X_{t_{k+1}} \mid X_{t_k}, \theta)} \prod_{k=0}^{m-2}\frac{\pi(X_{t_{k+1}} \mid X_{t_k}, X_{\tau_{i+1}}, \theta)}{\pi(X^*_{t_{k+1}} \mid X^*_{t_k}, X_{\tau_{i+1}}, \theta)},$$

where $X^*_{t_0} = X_{t_0} = X_{\tau_i}$ and $X^*_{t_m} = X_{t_m} = X_{\tau_{i+1}}$. For the case where only one data point is imputed between two observations (i.e. $m = 2$) this reduces to

$$\zeta(X^{imp*}_{(\tau_i,\tau_{i+1})}, X^{imp}_{(\tau_i,\tau_{i+1})}) = 1 \wedge \frac{\pi(X^*_{t_1} \mid X_{\tau_i}, \theta)\pi(X_{\tau_{i+1}} \mid X^*_{t_1}, \theta)}{\pi(X_{t_1} \mid X_{\tau_i}, \theta)\pi(X_{\tau_{i+1}} \mid X_{t_1}, \theta)} \frac{\pi(X_{t_1} \mid X_{\tau_i}, X_{\tau_{i+1}}, \theta)}{\pi(X^*_{t_1} \mid X_{\tau_i}, X_{\tau_{i+1}}, \theta)}$$

$$= 1 \wedge \left[\frac{\pi(X^*_{t_1} \mid X_{\tau_i}, \theta)\pi(X_{\tau_{i+1}} \mid X^*_{t_1}, \theta)}{\pi(X_{t_1} \mid X_{\tau_i}, \theta)\pi(X_{\tau_{i+1}} \mid X_{t_1}, \theta)}\right.$$

$$\left.\frac{\pi(X_{t_1} \mid X_{\tau_i}, \theta)\,\pi(X_{\tau_{i+1}} \mid X_{t_1}, \theta)/\pi(X_{\tau_{i+1}} \mid X_{\tau_i}, \theta)}{\pi(X^*_{t_1} \mid X_{\tau_i}, \theta)\,\pi(X_{\tau_{i+1}} \mid X^*_{t_1}, \theta)/\pi(X_{\tau_{i+1}} \mid X_{\tau_i}, \theta)}\right]$$

$$= 1.$$

This relation holds for any (approximated) transition density $\pi(X_{t_{k+1}} \mid X_{t_k}, \theta)$.

# Appendix C. Choice of path update interval

For choosing the update interval, we use the random block size algorithm as suggested in [3]. Assuming that the augmented path contains a total of $n + 1$ data points $Y_0, \ldots, Y_n$, it is divided into update segments $Y_{(c_0, c_1)}, Y_{(c_1, c_2)}, \ldots$ by the following algorithm:

(1) Set $c_0 = 0$ and $j = 1$.
(2) While $c_{j-1} < n$:
 (a) Draw $Z \sim \text{Po}(\lambda)$ and set $c_j = \min\{c_{j-1} + Z, n\}$.
 (b) Increment $j$.

Here, $Z \sim \text{Po}(\lambda)$ denotes the Poisson distribution with parameter $\lambda$.

Such a random choice of the path update interval is a simple way to vary the set of points that are updated together within one iteration.

# Appendix D. Additional example: Cox–Ingersoll–Ross process

The one-dimensional CIR process fulfils the SDE

$$\mathrm{d}X_t = \alpha(\beta - X_t)\,\mathrm{d}t + \sigma\sqrt{X_t}\,\mathrm{d}B_t, \quad X_0 = x_0,$$

with starting value $x_0 \in \mathbb{R}_+$ and parameters $\alpha, \beta, \sigma \in \mathbb{R}_+$. If $2\alpha\beta > \sigma^2$, the process is strictly positive (i.e. $\mathcal{X} = \mathbb{R}_+$) otherwise it is non-negative (i.e. $\mathcal{X} = \mathbb{R}_0$). The transition density is explicitly known as

$$p(s, x, t, y) = c\left(\frac{v}{u}\right)^{\eta/2} \mathrm{e}^{-(u+v)} I_\eta(2\sqrt{uv})$$

for $t > s \geq 0$, where

$$c = \frac{2\alpha}{\sigma^2(1 - \mathrm{e}^{-\alpha(t-s)})}, \quad u = cx\,\mathrm{e}^{-\alpha(t-s)}, \quad v = cy, \quad \eta = \frac{2\alpha\beta}{\sigma^2} - 1,$$

and $I_\eta$ denotes the modified Bessel function of the first kind of order $\eta$, i.e.

$$I_\eta(z) = \sum_{k=0}^{\infty} \left(\frac{z}{2}\right)^{2k+\eta} \frac{1}{k!\,\Gamma(k + \eta + 1)}$$

for $z \in \mathbb{R}$, where $\Gamma$ is the Gamma function.

For the CIR process, we have $\sigma(X_t, \theta) = \sigma\sqrt{X_t}$ with parameter $\sigma > 0$, the process taking values in $\mathbb{R}_0$. We therefore obtain a lower bound for the possible values of $X_{t_{k+1}}$ when applying the Milstein scheme

$$X_{t_{k+1}} \geq \left(\alpha(\beta - X_{t_k}) - \frac{1}{4}\sigma^2\right)\Delta t_k =: l_{\text{left}}.$$

The second bound that occurs when combining the MB proposal with the Milstein scheme is as follows:

$$X_{t_{k+1}} \geq \beta - \frac{1}{\alpha}\left(\frac{1}{\Delta_+}X_{t_m} + \frac{1}{4}\sigma^2\right) =: l_{\text{right}}.$$

The set $\mathcal{D}$ of feasible points of $X_{t_{k+1}}$ for the CIR process when combining the MB proposal with the Milstein scheme is thus $\mathcal{D} = [l, \infty)$ with $l := \max(0, l_{\text{left}}, l_{\text{right}})$.

For the simulation study, we generated 100 paths of the CIR process in the time interval $[0, 1]$ with the parameter combination $\theta = (\alpha, \beta, \sigma^2)^{\mathrm{T}} = (1, 1, 2)^{\mathrm{T}}$ and initial value $x_0 = 10$. From each path, we took 20 equidistant points and ran each of the described estimation methods once for 1 h to perform inference for the parameters $\beta$ and $\sigma^2$, assuming $\alpha$ to be known. For the prior distribution of the parameters, we assumed that they were independently distributed with $\beta \sim \text{IG}(\kappa_b = 3, \nu_b = 3)$ and $\sigma^2 \sim \text{IG}(\kappa_s = 3, \nu_s = 4)$. The *a priori* expectations of the parameters are thus $\mathbb{E}(\beta) = \frac{3}{2}$ and $\mathbb{E}(\sigma^2) = 2$. For each estimation procedure, the steps as outlined in §5 were taken. As proposal densities for the parameters in Step (3), we used $\beta^* \sim \mathcal{LN}(\log \beta_{i-1}, 0.25)$ and $\sigma^{2*} \sim \mathcal{LN}(\log \sigma^2_{i-1}, 0.25)$.

The sampling results are summarized in figures 10 and 11 and tables 3 and 4. Similar to the results for the GBM, the use of the Milstein approximation does not consistently improve the overall accuracy for the drift parameter $\beta$. The accuracy increases for increasing $m$ for most of the methods. Only combination MBM-M has lower accuracy for $m = 5$ due to the low sampling efficiency and the

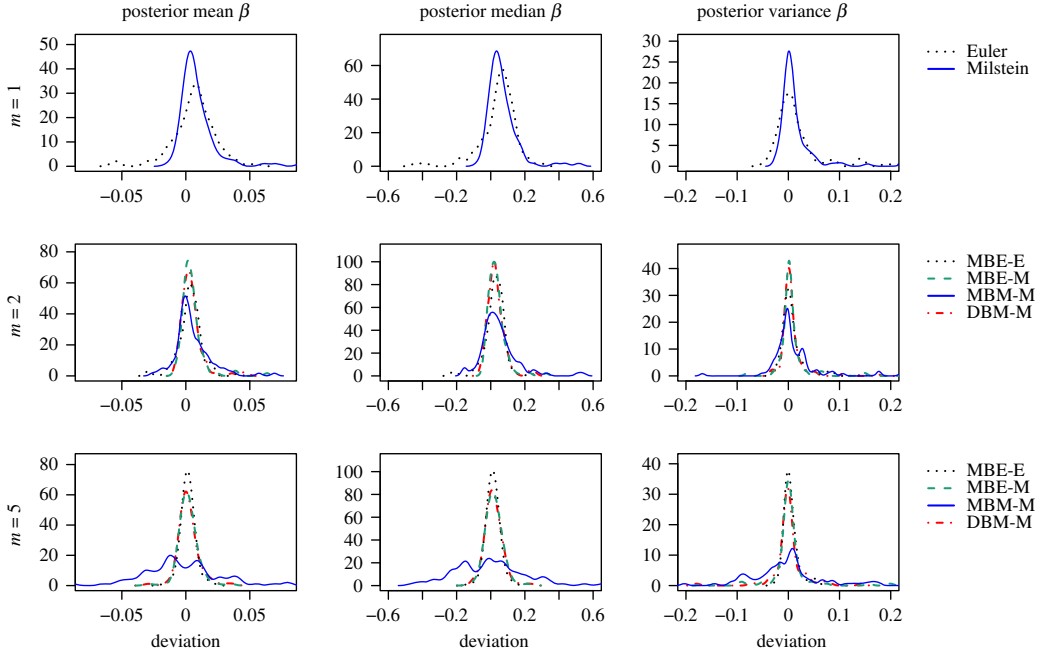

**Figure 10.** Sampling results for $\beta$ obtained by each of the estimation procedures. Each density plot aggregates 100 deviations between the respective statistics (left: mean, middle: median, right: variance) calculated for the sample from the approximated posterior and for the sample from the true posterior distribution, one for each of the 100 sample paths of the CIR process. The rows show results for different numbers $m$ of subintervals between two observations. For $m = 1$, no data points were imputed and only Step (1) from §3, the parameter update, was repeated in the estimation procedure.

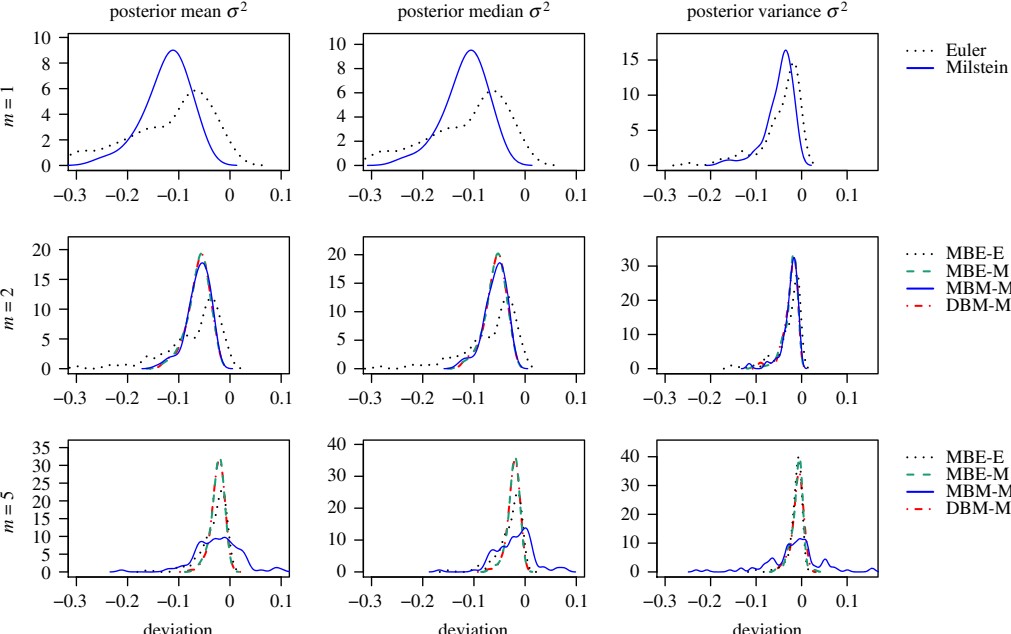

**Figure 11.** Sampling results for $\sigma^2$ as described in figure 10.

resulting low ESS. For the diffusion parameter $\sigma^2$, the use of the Milstein approximation and increasing $m$ both improve the overall accuracy. Again combination DBM-M achieves the highest accuracy, closely followed by MBE-M.

**Table 3.** Empirical characteristics for evaluating the overall accuracy of the parameter estimation procedures for different numbers $m$ of subintervals between two observations aggregated over 100 deviations between the respective statistics calculated for the sample from the approximated posterior and for the sample from the true posterior distribution, one for each of the 100 sample paths of the CIR process. The lowest RMSE per $m$ and per statistic is printed in bold.

| | method | RMSEs for $\beta$ | | | RMSEs for $\sigma^2$ | | |
| --- | --- | --- | --- | --- | --- | --- | --- |
| | | mean | median | variance | mean | median | variance |
| $m = 1$ | Euler | 0.0179 | 0.0115 | **0.0478** | 0.1603 | 0.1530 | 0.0673 |
| | Milstein | **0.0174** | **0.0110** | 0.0587 | **0.1306** | **0.1233** | **0.0595** |
| $m = 2$ | MBE-E | 0.0099 | 0.0064 | **0.0265** | 0.0910 | 0.0865 | 0.0417 |
| | MBE-M | 0.0105 | 0.0063 | 0.0413 | 0.0656 | 0.0619 | 0.0309 |
| | MBM-M | 0.0151 | 0.0120 | 0.0462 | 0.0658 | 0.0625 | 0.0325 |
| | DBM-M | **0.0097** | **0.0061** | 0.0330 | **0.0653** | **0.0617** | **0.0308** |
| $m = 5$ | MBE-E | **0.0052** | **0.0036** | 0.0144 | 0.0400 | 0.0380 | 0.0194 |
| | MBE-M | 0.0077 | 0.0049 | 0.0375 | 0.0271 | 0.0259 | **0.0156** |
| | MBM-M | 0.0307 | 0.0204 | 0.1103 | 0.0509 | 0.0420 | 0.0615 |
| | DBM-M | 0.0085 | 0.0052 | **0.0321** | 0.0270 | 0.0256 | 0.0156 |

**Table 4.** Empirical characteristics for evaluating the computational efficiency of the parameter estimation procedures for different numbers $m$ of subintervals between two observations aggregated over 100 trajectories of the CIR process. Each of the procedures was run for 1 h. Acceptance rates are defined to take values between 0 and 1. For $m = 1$, no data points were imputed and only Step (1) from §3, the parameter update, was repeated in the estimation procedure. Specifications for the computing power are stated in the main text. c.v. denotes the coefficient of variation.

| | method | number of iterations after 1 h | | multivariate effective sample size | | acceptance rate of the parameters | | acceptance rate of the path | |
| --- | --- | --- | --- | --- | --- | --- | --- | --- | --- |
| | | mean | c.v. | mean | c.v. | mean | c.v. | mean | c.v. |
| $m = 1$ | Euler | 23 461 023 | 0.11 | 2 422 521 | 0.14 | 0.443 | 0.03 | — | — |
| | Milstein | 4 685 450 | 0.03 | 480 549 | 0.08 | 0.442 | 0.03 | — | — |
| $m = 2$ | MBE-E | 8 482 241 | 0.06 | 422 034 | 0.10 | 0.384 | 0.03 | 0.964 | 0.01 |
| | MBE-M | 1 944 229 | 0.05 | 94 071 | 0.10 | 0.383 | 0.03 | 0.957 | 0.01 |
| | MBM-M | 186 588 | 0.06 | 9429 | 0.13 | 0.383 | 0.03 | 1.000 | 0.00 |
| | DBM-M | 1 905 354 | 0.04 | 95 262 | 0.10 | 0.383 | 0.03 | 0.968 | 0.01 |
| $m = 5$ | MBE-E | 6 851 197 | 0.05 | 114 344 | 0.10 | 0.272 | 0.03 | 0.976 | 0.01 |
| | MBE-M | 966 579 | 0.04 | 15 599 | 0.13 | 0.272 | 0.03 | 0.965 | 0.01 |
| | MBM-M | 37 648 | 0.12 | 574 | 0.25 | 0.272 | 0.03 | 0.993 | 0.00 |
| | DBM-M | 906 791 | 0.08 | 14 881 | 0.14 | 0.272 | 0.03 | 0.975 | 0.01 |

Also for the CIR process, the number of iterations completed after 1 h and the multivariate ESS of the obtained sample vary substantially between the different procedures. Both quantities are highest for combination MBE-E, they are similar for MBE-M and DBM-M, and particularly low for MBM-M.

# Appendix E. Analysis of the correlation between the parameters

In this section, we provide several plots (see figures 12–15) showing that the parameters of the two benchmark models are not strongly correlated in order to justify our use of independent parameter proposals in the simulation study.

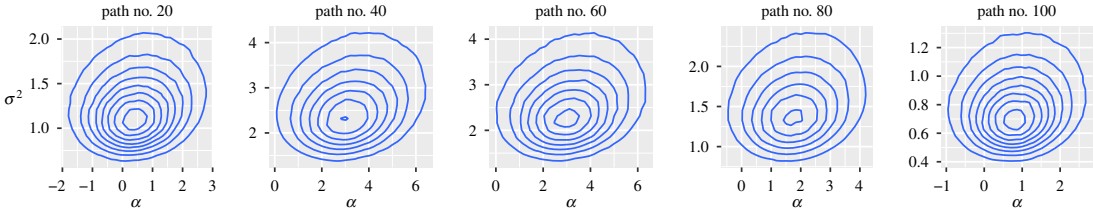

**Figure 12.** Two-dimensional density plots of the parameter samples from the true posterior distribution for exemplary paths of the GBM.

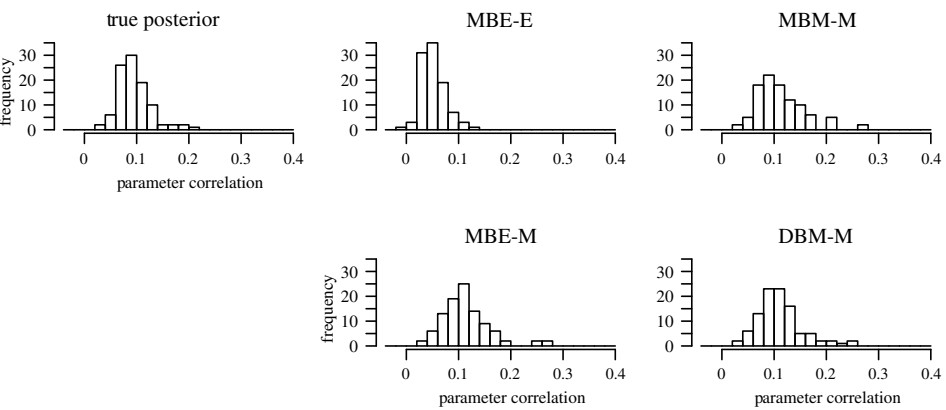

**Figure 13.** Histograms of the values of Pearson's correlation coefficient calculated for each of the 100 sample paths of the GBM for the parameter samples from the true posterior distributions and the parameter samples from the approximated posterior distributions obtained with one of the four considered methods for $m = 5$.

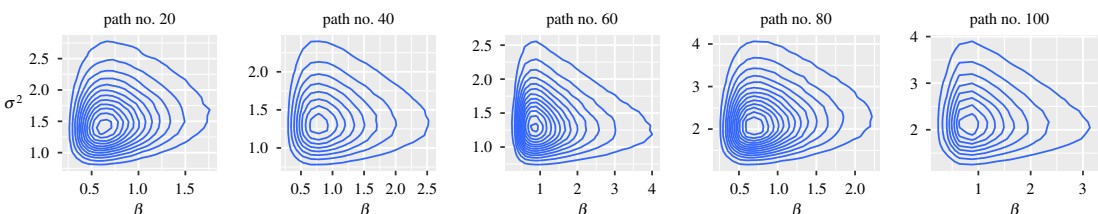

**Figure 14.** Two-dimensional density plots of the parameter samples from the true posterior distribution for exemplary paths of the CIR process.

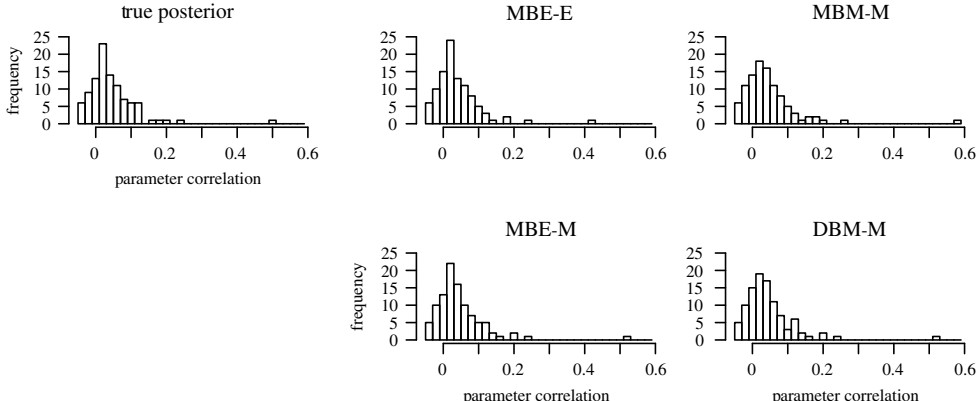

**Figure 15.** Histograms of the values of Pearson's correlation coefficient calculated for each of the 100 sample paths of the CIR process for the parameter samples from the true posterior distributions and the parameter samples from the approximated posterior distributions obtained with one of the four considered methods for $m = 5$.

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
