## [Reviewer comments · Royal Society Open Science]

Review History

RSOS-200270.R0 (Original submission)

Review form: Reviewer 1

Is the manuscript scientifically sound in its present form?

No

Are the interpretations and conclusions justified by the results?

No

Is the language acceptable?

Yes

Do you have any ethical concerns with this paper?

No

Have you any concerns about statistical analyses in this paper?

Yes

Recommendation?

Major revision is needed (please make suggestions in comments)

Comments to the Author(s)

Please see the attached pdf (Appendix A).

Review form: Reviewer 2**Is the manuscript scientifically sound in its present form?**

Yes

Are the interpretations and conclusions justified by the results?

Yes

Is the language acceptable?

Yes

Do you have any ethical concerns with this paper?

No

Have you any concerns about statistical analyses in this paper?

Yes

Recommendation?

Major revision is needed (please make suggestions in comments)

Comments to the Author(s)

The paper compares the Euler-Maruyama and Milstein approximations to the transition density of a stochastic differential equation. The authors compare the approximations using two data augmentation Markov chain Monte Carlo algorithms. The first uses the (approximate) transition density to propose new points and the second uses the modified diffusion bridge. An alternate derivation of the modified diffusion bridge using the Milstein approximation is provided. The authors test all combinations of the two approximations (both for parameter updates and proposals) on the two data augmentation approaches, which leads to 8 different algorithms. They apply these algorithms on a geometric Brownian motion SDE as well as a Cox-Ingersoll-Ross example in Appendix D. The paper was interesting to read and the results were quite surprising. However, there are a few issues that should be addressed.

Main comments

As I understand it, the lower bound for the Milstein scheme disallows feasible values of X . In the Milstein MB proposal, no values below the lower bound are sampled and when the Milstein approximation to the transition density is used, values below the lower bound are given 0 weight. Is this correct?

How is the lower bound affected by the level of discretisation? A comparison of the exact transition density and the Euler-Maruyama and Milstein approximations is shown in Figure 2. In Figure 2b, the error in the Euler-Maruyama approximation can be made arbitrarily small by decreasing the time step. Can the same be said for the Milstein approximation in that example (in the context of the lower bound)?

If the lower bound cuts off feasible values, won't this bias the results? In practice, how does the lower bound affect inference results? In the simulation study in section 5, are the marginal posteriors for θ the same for all 8 methods?

Is there some intuition about why the Milstein approximation does not perform better than the Euler-Maruyama in the simulation study?

In the GBM example, the Euler-Maruyama and Milstein approximations give very similar results. In general, the Milstein scheme yields a more accurate approximation to the transition density than the Euler-Maruyama approximation. However, if the latter is already a good approximation, then the Milstein scheme would not necessarily improve the results.

Related to the above point. In Figure 2b, the Euler-Maruyama approximation with a time step of 0.1 is not very accurate, but in the simulation study it yields quite good results. The time step used in the simulation study (with no imputation) is 0.02 however. How accurate is the Euler-Maruyama approximation in this case? Please give a comparison/measure of the accuracy of the two approximations for the simulation study in section 5.

In the simulation study, 50 equidistant points are taken from each path. Are the results of the comparison the same when less points are used (e.g. 10 or 20)?

Minor comments

In Section 2.1, the stochastic Taylor expansion of the process is not defined.

Typo in the first equation in Section 2.2

Decision letter (RSOS-200270.R0)

14-Apr-2020

Dear Ms Pieschner,

The editors assigned to your paper ("Bayesian inference for diffusion processes: using higher-order approximations for transition densities") have now received comments from reviewers. We would like you to revise your paper in accordance with the referee and Associate Editor suggestions which can be found below (not including confidential reports to the Editor). Please note this decision does not guarantee eventual acceptance.

Please submit a copy of your revised paper before 07-May-2020. Please note that the revision deadline will expire at 00.00am on this date. If we do not hear from you within this time then it will be assumed that the paper has been withdrawn. In exceptional circumstances, extensions may be possible if agreed with the Editorial Office in advance. We do not allow multiple rounds of revision so we urge you to make every effort to fully address all of the comments at this stage. If deemed necessary by the Editors, your manuscript will be sent back to one or more of the original reviewers for assessment. If the original reviewers are not available, we may invite new reviewers.

When submitting your revised manuscript, you must respond to the comments made by the referees and upload a file "Response to Referees" in "Section 6 - File Upload". Please use this to document how you have responded to the comments, and the adjustments you have made. In

order to expedite the processing of the revised manuscript, please be as specific as possible in your response.

- Data accessibility

If you wish to submit your supporting data or code to Dryad (<http://datadryad.org/>), or modify your current submission to dryad, please use the following link:
<http://datadryad.org/submit?journalID=RSOS&manu=RSOS-200270>

- Competing interests

- Authors' contributions

- Acknowledgements

- Funding statement

Once again, thank you for submitting your manuscript to Royal Society Open Science and I look

forward to receiving your revision. If you have any questions at all, please do not hesitate to get in touch.

Kind regards,

Andrew Dunn

on behalf of Professor Andreas Kyprianou (Associate Editor) and Mark Chaplain (Subject Editor)

Associate Editor's comments (Professor Andreas Kyprianou):

Associate Editor: 1

Comments to the Author:

We have two good reports from the referees. They have both raised a number of points which need addressing before we can move to acceptance. There is quite a lot to address, but even if no action is needed after meditation on some of the points, please prepare a careful response.

Comments to Author:

Reviewers' Comments to Author:

Reviewer: 1

Comments to the Author(s)

Please see the attached pdf.

Reviewer: 2

Comments to the Author(s)

The paper compares the Euler-Maruyama and Milstein approximations to the transition density of a stochastic differential equation. The authors compare the approximations using two data augmentation Markov chain Monte Carlo algorithms. The first uses the (approximate) transition density to propose new points and the second uses the modified diffusion bridge. An alternate derivation of the modified diffusion bridge using the Milstein approximation is provided. The authors test all combinations of the two approximations (both for parameter updates and proposals) on the two data augmentation approaches, which leads to 8 different algorithms. They apply these algorithms on a geometric Brownian motion SDE as well as a Cox-Ingersoll-Ross example in Appendix D. The paper was interesting to read and the results were quite surprising. However, there are a few issues that should be addressed.

Main comments

As I understand it, the lower bound for the Milstein scheme disallows feasible values of X . In the Milstein MB proposal, no values below the lower bound are sampled and when the Milstein approximation to the transition density is used, values below the lower bound are given 0 weight. Is this correct?

How is the lower bound affected by the level of discretisation? A comparison of the exact transition density and the Euler-Maruyama and Milstein approximations is shown in Figure 2. In Figure 2b, the error in the Euler-Maruyama approximation can be made arbitrarily small by decreasing the time step. Can the same be said for the Milstein approximation in that example (in the context of the lower bound)?

If the lower bound cuts off feasible values, won't this bias the results? In practice, how does the lower bound affect inference results? In the simulation study in section 5, are the marginal posteriors for θ the same for all 8 methods?

Is there some intuition about why the Milstein approximation does not perform better than the Euler-Maruyama in the simulation study?

In the GBM example, the Euler-Maruyama and Milstein approximations give very similar results. In general, the Milstein scheme yields a more accurate approximation to the transition density than the Euler-Maruyama approximation. However, if the latter is already a good approximation, then the Milstein scheme would not necessarily improve the results.

Related to the above point. In Figure 2b, the Euler-Maruyama approximation with a time step of 0.1 is not very accurate, but in the simulation study it yields quite good results. The time step used in the simulation study (with no imputation) is 0.02 however. How accurate is the Euler-Maruyama approximation in this case? Please give a comparison/measure of the accuracy of the two approximations for the simulation study in section 5.

In the simulation study, 50 equidistant points are taken from each path. Are the results of the comparison the same when less points are used (e.g. 10 or 20)?

Minor comments

In Section 2.1, the stochastic Taylor expansion of the process is not defined.
Typo in the first equation in Section 2.

Author's Response to Decision Letter for (RSOS-200270.R0)

See Appendix B.

RSOS-200270.R1 (Revision)

Review form: Reviewer 1

Is the manuscript scientifically sound in its present form?

Yes

Are the interpretations and conclusions justified by the results?

Yes

Is the language acceptable?

Yes

Do you have any ethical concerns with this paper?

No

Have you any concerns about statistical analyses in this paper?

No

Recommendation?

Accept with minor revision (please list in comments)

Comments to the Author(s)

This version of the manuscript really is a substantial improvement. The simulation study, in particular, is now much clearer and more informative. I have two "medium" comments and several minor comments.

(1) In the simulation study it would still be good to be clear between 2(a) and 2(b), Specifically, how much are the MSEs of the posterior quantities down to approximation error and how much down to not a high enough ESS. I suspect it's nearly all down to approximation error - but can you quantify this? e.g. for the MSE for the mean, you also have a typical size of the posterior variance, you know the ESS, so the contribution to the MSE from MCMC variance can be calculated. The remainder must be bias^2 . Is one much larger than the other?

(2) You have different acceptance rates for the different m values, and are reporting this. Usually one tunes a RWM (of log RWM) algorithm to obtain a particular acceptance rate (around 40% in dimension 1, I think). The different acceptance rates are actually suggesting that the tunings may not be comparable - different scalings could have been better for different m values. This needs mentioning. However, I'm not suggesting you redo the simulation study; instead you could, for example, refer to the Efficiency vs acceptance rate curve in Fig 3 of Roberts and Rosenthal (2001), Stat. Sci., which shows that (at least in the high-dimensional limit) efficiency is pretty robust to changes in the acceptance rate of around 0.1 either side of the optimum. So your ESS's are probably comparable to within a factor of two, or even tighter.

MINOR

P17 the first paragraph talks about a sample from the true posterior, but it's not until the third paragraph that this is described. Perhaps add "(see below)" or similar to the first paragraph? p17 "obtain a two-dimensional Markov chain". Pedantic, but strictly taking (α, σ^2) from all your MCMC schemes that also involve imputed X values, the set of these two parameter values is not Markovian. Please use a different description.

P18 100 of such -> a hundred such

P19 data is a plural word, so "when data are imputed" and "data are introduced"

P21 "This strength [of the Euler scheme] is weakened by the limited applicability of the Milstein approximation." I cannot make sense of this statement.

P21 Less iterations -> Fewer iterations

P22 It was expected does make -> It was expected ... would make

Review form: Reviewer 2

Is the manuscript scientifically sound in its present form?

Yes

Are the interpretations and conclusions justified by the results?

Yes

Is the language acceptable?

Yes

Do you have any ethical concerns with this paper?

No

Have you any concerns about statistical analyses in this paper?

No

Recommendation?

Accept with minor revision (please list in comments)

Comments to the Author(s)

The authors have done a great job with the revision. I only have one comment regarding the DBM proposal function.

1. In the simulation studies, the DBM-M and MBE-E methods give the best overall accuracy for the diffusion and drift coefficients, respectively. In the latter case, the computational efficiency of the Euler scheme outweighs any increase in accuracy provided by the Milstein scheme when approximating the transition density. Moreover, the DBM-M combination generally seems to outperform the MBE-M combination in terms of overall accuracy and multivariate ESS. Considering this, I believe the combination DBM-E would also be of interest. The results from this combination (especially when compared to MBE-E) would give an indication of the usefulness of the Milstein scheme (or other higher-order approximations) for proposal functions. Even though DBM is not as generally applicable as MBE, it might be an interesting area of future research.
2. Typo in Equation (13): τ_{-1} is used in the product instead of τ_{-i}

Review form: Reviewer 3

Is the manuscript scientifically sound in its present form?

Yes

Are the interpretations and conclusions justified by the results?

Yes

Is the language acceptable?

Yes

Do you have any ethical concerns with this paper?

No

Have you any concerns about statistical analyses in this paper?

No

Recommendation?

Accept with minor revision (please list in comments)

Comments to the Author(s)

The paper treats a relevant question - in fact, I am slightly surprised that this has not been addressed before in the literature although the reason may be that the answer is largely negative: few additional gains are to be made from using the Milstein scheme over an Euler scheme in Bayesian parameter estimation for diffusion processes.

Other than some minor comments (mostly typographical or stylistic but also one point where I believe the paper is mathematically incorrect), I feel that the paper, now in its second round, is essentially ready for publication.

p3 l38: Maybe it would be good to point to the CIR example in appendix here already. I accept the authors' argument about not reversing the position of CIR vs GBM in appendix vs paper in the interest of having as accessible an example as possible in the paper. Nonetheless, the previous criticism that many higher order terms in the stochastic Taylor expansion are zero in the GBM case stands and it is therefore important to make clear that the content applies to other SDEs where this is not the case, too.

p4 l 43 "stochastic Taylor expansion of the process $(X_t)_{t \geq 0}$, and has strong order of convergence 0.5." would be my suggested phrasing

p4 145 The notation \mathbf{Y} is a bit problematic: while fairly intuitive, it seems to be neither defined nor used at all elsewhere in the paper. Also, it once arises with the superscript Δ and once without.

p5 138 claims that the equality at line 35 following "More generally, we require that" (eliminating those stochastic double integrals that cannot be computed analytically from the Milstein approximation) holds if and only if σ has at most one non-zero entry per row. I believe that this condition is sufficient but not necessary. I believe the undesirable double stochastic integrals also disappear if σ is constant but full. Another counter example is the matrix $(2+\cos(x), 0; 1, 1)$ (first row with entries $2+\cos(x)$ on diagonal and zero off diagonal, second row has both entries equal to one). This makes the Milstein scheme slightly more widely applicable than claimed by the authors but overall the conditions are still rather restrictive, so I don't think much else needs to change in the paper.

p7 115 "denotes the multivariate Gaussian density with mean $a \in \mathbb{R}^d$ and covariance matrix $b \in \mathbb{R}^d \times d$ evaluated at y ." with my suggested addition for clarity.

p9 148 you claim that " n denotes the total number of data points in the augmented path" but it seems to me that, while there are n terms in the product in the displayed formula preceding this sentence, these n terms actually require $X_{\{t_0\}}$ to $X_{\{t_n\}}$ to compute, so that we have $n+1$ data points in total.

p20, Figure 8 : Green lines are hard to see here. Also, why is the posterior variance of α almost always underestimated? This is concerning in the application as it will lead to credible intervals that are too narrow and therefore deserves comment.

p21 reads "One of the strengths of the original (Euler-based) MCMC scheme is its generic character and applicability. Through this, it possesses a practical advantage over otherwise more sophisticated methods such as the Exact Algorithm ([27]). This strength, however, is weakened by the limited applicability of the Milstein approximation especially in the multidimensional setting." I find this confusing. You are saying that the Euler-scheme's strength of generic character and applicability is weakened by the Milstein (not the Euler) approximation's limited applicability.

p21 "Less iterations are completed for the methods involving the Milstein scheme and also the ESS is substantially lower." should probably read "Fewer iterations..."

Figures D1 and D2 have a similar visibility problem with the dashed green lines.

Overall, apart from my one mathematical point querying whether the condition specified for all non-analytically computable double stochastic integrals to disappear is necessary, the content seems mathematically and algorithmically correct to me and I commend the authors for thoroughly and methodically investigating a natural open question and writing up this investigation properly even though the results do not lend themselves to making spectacular claims on improvements.

Decision letter (RSOS-200270.R1)

Dear Ms Pieschner,

On behalf of the Editors, we are pleased to inform you that your Manuscript RSOS-200270.R1 "Bayesian inference for diffusion processes: using higher-order approximations for transition densities" has been accepted for publication in Royal Society Open Science subject to minor revision in accordance with the referees' reports. Please find the referees' comments along with any feedback from the Editors below my signature.

Please submit your revised manuscript and required files (see below) no later than 7 days from today's (ie 08-Sep-2020) date. Note: the ScholarOne system will 'lock' if submission of the revision is attempted 7 or more days after the deadline. If you do not think you will be able to meet this deadline please contact the editorial office immediately.

Kind regards,
Lianne Parkhouse
Editorial Coordinator
Royal Society Open Science
openscience@royalsociety.org

on behalf of Professor Andreas Kyprianou (Associate Editor) and Mark Chaplain (Subject Editor)
openscience@royalsociety.org

Associate Editor Comments to Author (Professor Andreas Kyprianou):

There are still quite a few things that the referees have highlighted but they are all willing to allow the corrections in good faith and proceed to accept with minor revisions. I am recommending accordingly.

Reviewer comments to Author:

Reviewer: 1

Comments to the Author(s)

This version of the manuscript really is a substantial improvement. The simulation study, in particular, is now much clearer and more informative. I have two "medium" comments and several minor comments.

(1) In the simulation study it would still be good to be clear between 2(a) and 2(b), Specifically, how much are the MSEs of the posterior quantities down to approximation error and how much down to not a high enough ESS. I suspect it's nearly all down to approximation error - but can you quantify this? e.g. for the MSE for the mean, you also have a typical size of the posterior variance, you know the ESS, so the contribution to the MSE from MCMC variance can be calculated. The remainder must be bias^2 . Is one much larger than the other?

(2) You have different acceptance rates for the different m values, and are reporting this. Usually one tunes a RWM (of log RWM) algorithm to obtain a particular acceptance rate (around 40% in dimension 1, I think). The different acceptance rates are actually suggesting that the tunings may not be comparable - different scalings could have been better for different m values. This needs mentioning. However, I'm not suggesting you redo the simulation study; instead you could, for example, refer to the Efficiency vs acceptance rate curve in Fig 3 of Roberts and Rosenthal (2001), Stat. Sci., which shows that (at least in the high-dimensional limit) efficiency is pretty robust to changes in the acceptance rate of around 0.1 either side of the optimum. So you're ESS's are probably comparable to within a factor of two, or even tighter.

MINOR

P17 the first paragraph talks about a sample from the true posterior, but it's not until the third paragraph that this is described. Perhaps add "(see below)" or similar to the first paragraph? p17 "obtain a two-dimensional Markov chain". Pedantic, but strictly taking (α, σ^2) from all your MCMC schemes that also involve imputed X values, the set of these two parameter values is not Markovian. Please use a different description.

P18 100 of such -> a hundred such

P19 data is a plural word, so "when data are imputed" and "data are introduced"

P21 "This strength [of the Euler scheme] is weakened by the limited applicability of the Milstein approximation." I cannot make sense of this statement.

P21 Less iterations -> Fewer iterations

P22 It was expected does make -> It was expected ... would make

Reviewer: 2

Comments to the Author(s)

The authors have done a great job with the revision. I only have one comment regarding the DBM proposal function.

1. In the simulation studies, the DBM-M and MBE-E methods give the best overall accuracy for the diffusion and drift coefficients, respectively. In the latter case, the computational efficiency of the Euler scheme outweighs any increase in accuracy provided by the Milstein scheme when approximating the transition density. Moreover, the DBM-M combination generally seems to outperform the MBE-M combination in terms of overall accuracy and multivariate ESS. Considering this, I believe the combination DBM-E would also be of interest. The results from this combination (especially when compared to MBE-E) would give an indication of the usefulness of the Milstein scheme (or other higher-order approximations) for proposal functions. Even though DBM is not as generally applicable as MBE, it might be an interesting area of future research.

2. Typo in Equation (13): τ_{-1} is used in the product instead of τ_{-i}

Reviewer: 3

Comments to the Author(s)

The paper treats a relevant question - in fact, I am slightly surprised that this has not been addressed before in the literature although the reason may be that the answer is largely negative: few additional gains are to be made from using the Milstein scheme over an Euler scheme in Bayesian parameter estimation for diffusion processes.

Other than some minor comments (mostly typographical or stylistic but also one point where I believe the paper is mathematically incorrect), I feel that the paper, now in its second round, is essentially ready for publication.

p3 138: Maybe it would be good to point to the CIR example in appendix here already. I accept the authors' argument about not reversing the position of CIR vs GBM in appendix vs paper in the interest of having as accessible an example as possible in the paper. Nonetheless, the previous criticism that many higher order terms in the stochastic Taylor expansion are zero in the GBM case stands and it is therefore important to make clear that the content applies to other SDEs where this is not the case, too.

p4 143 "stochastic Taylor expansion of the process $(X_t)_{t \geq 0}$, and has strong order of convergence 0.5." would be my suggested phrasing

p4 145 The notation \mathbf{Y} is a bit problematic: while fairly intuitive, it seems to be neither defined nor used at all elsewhere in the paper. Also, it once arises with the superscript Δ and once without.

p5 138 claims that the equality at line 35 following "More generally, we require that" (eliminating those stochastic double integrals that cannot be computed analytically from the Milstein approximation) holds if and only if σ has at most one non-zero entry per row. I believe that this condition is sufficient but not necessary. I believe the undesirable double stochastic integrals also disappear if σ is constant but full. Another counter example is the matrix $(2+\cos(x), 0; 1, 1)$ (first row with entries $2+\cos(x)$ on diagonal and zero off diagonal, second row has both entries equal to one). This makes the Milstein scheme slightly more widely applicable than claimed by the authors but overall the conditions are still rather restrictive, so I don't think much else needs to change in the paper.

p7 115 "denotes the multivariate Gaussian density with mean $a \in \mathbb{R}^d$ and covariance matrix $b \in \mathbb{R}^d \times d$ evaluated at y ." with my suggested addition for clarity.

p9 148 you claim that " n denotes the total number of data points in the augmented path" but it seems to me that, while there are n terms in the product in the displayed formula preceding this sentence, these n terms actually require $X_{\{t_0\}}$ to $X_{\{t_n\}}$ to compute, so that we have $n+1$ data points in total.

p20, Figure 8 : Green lines are hard to see here. Also, why is the posterior variance of α almost always underestimated? This is concerning in the application as it will lead to credible intervals that are too narrow and therefore deserves comment.

p21 reads "One of the strengths of the original (Euler-based) MCMC scheme is its generic character and applicability. Through this, it possesses a practical advantage over otherwise more sophisticated methods such as the Exact Algorithm ([27]). This strength, however, is weakened by the limited applicability of the Milstein approximation especially in the multidimensional setting." I find this confusing. You are saying that the Euler-scheme's strength of generic character and applicability is weakened by the Milstein (not the Euler) approximation's limited applicability.

p21 "Less iterations are completed for the methods involving the Milstein scheme and also the ESS is substantially lower." should probably read "Fewer iterations..."

Figures D1 and D2 have a similar visibility problem with the dashed green lines.

Overall, apart from my one mathematical point querying whether the condition specified for all non-analytically computable double stochastic integrals to disappear is necessary, the content seems mathematically and algorithmically correct to me and I commend the authors for thoroughly and methodically investigating a natural open question and writing up this investigation properly even though the results do not lend themselves to making spectacular claims on improvements.

===PREPARING YOUR MANUSCRIPT===

===PREPARING YOUR REVISION IN SCHOLARONE===

-- If you have uploaded ESM files, please ensure you follow the guidance at <https://royalsociety.org/journals/authors/author-guidelines/#supplementary-material> to include a suitable title and informative caption. An example of appropriate titling and captioning may be found at https://figshare.com/articles/Table_S2_from_Is_there_a_trade-off_between_peak_performance_and_performance_breadth_across_temperatures_for_aerobic_scops_in_teleost_fishes_/3843624.

Author's Response to Decision Letter for (RSOS-200270.R1)

See Appendix C.

Decision letter (RSOS-200270.R2)

Dear Ms Pieschner,

It is a pleasure to accept your manuscript entitled "Bayesian inference for diffusion processes: using higher-order approximations for transition densities" in its current form for publication in Royal Society Open Science.

You can expect to receive a proof of your article in the near future. Please contact the editorial office (openscience_proofs@royalsociety.org) and the production office (openscience@royalsociety.org) to let us know if you are likely to be away from e-mail contact -- if

you are going to be away, please nominate a co-author (if available) to manage the proofing process, and ensure they are copied into your email to the journal.

on behalf of Professor Andreas Kyprianou (Associate Editor) and Mark Chaplain (Subject Editor)
openscience@royalsociety.org

Appendix A

Review of: Bayesian inference for diffusion processes: using higher-order approximations for transition densities (RSOS-200270).

Summary and main points

Summary Inference on the parameters, and potentially the paths, of SDEs is typically performed by approximating the paths through a discretisation and then using the Euler-Maruyama method to estimate the likelihood. Typically the Euler-Maruyama (E-M) method is also used to create the proposed paths, whether through forward simulations or some kind of conditioning on future observations. The main exception to this is in those rare cases (such as in one dimension) where the Exact Algorithm or its evolutions can be used to perform exact inference. This article examines replacing the E-M method by the Milstein method, which contains an additional ‘correction’ term and has a higher order of accuracy. The use of the method is described, including the myriad difficulties and problems, and then the method is tested on the 1D example of Geometric Brownian Motion (GBM) [and the CIR process in the appendix]. Little if any improvement in accuracy is found, and the computational cost is much higher than for the E-M method.

Main points This has the potential to be an interesting paper. To my knowledge, no-one else has looked at the question of replacing E-M with Milstein, yet it is a natural question. This paper starts to answer it, and describes the fundamental issues well, but needs much clearer and better thought-out simulation studies.

0. I certainly had not appreciated the difficulties with using the Milstein scheme in dimension > 1 , and this is clearly pointed out in the paper. Further, even in one dimension, the Milstein scheme has an odd property that because it is quadratic in ΔB the increment has either an upper or lower bound, and this follows through to issues with the likelihood. This is discussed in the paper, though it might be a clearer connection can be made between the scheme itself and the likelihood issues (see medium point 3). It also seems to be that the set of cases where the Milstein scheme can be used is only slightly larger than the set for which the EA is valid, so at least a brief discussion of why one might prefer Milstein seems warranted.
1. I am missing the motivation to use the Milstein scheme for proposing a path and the Euler scheme for evaluating the likelihood, or vice versa. Why would this be a consistent, sensible approach? Also, it would be good to have confirmation that if the Milstein scheme is used to propose a path then the q for that path, needed in the acceptance ratio, is also calculated using the Milstein scheme; this has to happen for consistency, and also surely would potentially lead to issues when simulations are by Euler yet likelihood by Milstein that certain proposals would have a proposal probability of zero because of point 0 above. Unless there is a good justification that I’m missing, I would suggest just two scenarios here, rather than four: Milstein for everything or E-M for everything.
2. There are two factors being confounded. (a) The accuracy of the approximation to the posterior distribution given a particular discretisation (m) and SDE integration scheme (Euler/Milstein), and (b) the accuracy with which that approximate posterior distribution has been obtained by the MCMC; the mixing of the MCMC depends on the choice of bridge scheme: left-conditioned (i.e., no bridge) or MB. Also, with the MCMC scheme used, the mixing degrades with m . Interest might be purely in (a) but you need to control for (b) or you might be interested in the overall best accuracy obtainable given a fixed amount of computer time. I think both would be of interest, but from the way the results are presented it seems like only the latter is considered, and even then, this is not made explicit. To consider the former you would need to ensure that all schemes are run for long enough so that that you are confident any differences in inference (e.g. posterior means) arise from (a). This is straightforward to ensure given e.g. the ESSs and the posterior variances.
3. The wrong comparisons are being made. Currently 100 data sets are simulated and the distribution of posterior means under the different schemes are compared with the distribution of the true MAP estimates and the MLEs. This (a) *is not comparing like with like* and it (b) *is not using the fact that the same 100 data sets are used*. For (a) since GBM is tractable, you can obtain the true posterior distribution for the parameters (as you must have used to obtain the MAP), so run a *really long* MCMC

on the two parameters and find, to very high accuracy, the *true* posterior mean for each data set. For (b) you then pair the results from a particular approximate run on a particular data set with the truth for that same data set. You can then provide (i) density plots of these *discrepancies* (i.e. approximating the distribution, over data sets, of the error in the estimate of the posterior mean), and (ii) RMSEs for these quantities. Ideally, if your interest is in “accuracy after a given computational time” you would use the same computational time for all runs. Currently it is entirely unclear what Tables 1 and 2 depict; I would guess that what is reported as “bias” is actually the deviation from the true parameter value; similarly RMSE. Or perhaps they are looking at deviations from the MAP values? But this is not what you want to compare against, as already stated. The posterior for σ^2 , at least, is not symmetric so the MAP and posterior mean are not the same. I would also be tempted to look at posterior medians rather than (or in addition to) posterior means, since they are more robust. Ideally you would also look at estimates of the posterior variance.

4. The simulation study comparisons may have been doomed from the start. It may be that the inter-observation time is so small that there is very little difference between. E-M and Milstein as both are very accurate - and far more accurate than the MCMC is able to discern given the 10^5 iterations. One would expect to see this e.g. in Table 1 for $m=2$, where Milstein-Milstein should have a lower bias and RMSE than E-M-E-M. It does not. This may be down to the point being made here, or that made in 3. or both. At the moment I cannot tell.

4.(b) The left-conditioned scheme usually performs terribly when observations are exact (at least for $m \geq 5$). The fact that it does as well as the MB here suggests indeed that the observations are too close together to allow much freedom. Given that (as is well known and demonstrated in Figure 4) the left-conditioned scheme is usually poor in the case of exact observations, and given that these schemes only affect the efficiency of the MCMC, not the accuracy of the approximate posterior if the MCMC were carried on to ∞ , it would not be unreasonable to drop it the left-conditioned scheme. One alternative possibility would be to replace this bifurcation with two uses of the MB: as in [6] and within particle MCMC.

5. The MCMC scheme being used was state-of-the art 20 years ago. The mixing criticism is acknowledged in the article, but still, inference has moved on considerably since then. In terms of latent variable schemes similar to the one used, that in [6, Golightly and Wilkinson, CSDA] is hard to beat; however, nowadays the usual method is particle MCMC. I bring this point up for two reasons. If the aim is purely the one in 2(a) then you want the MCMC to mix as thoroughly as possible as quickly as possible, so unless it is feasible to increase the number of iterations of your current scheme until all the ESSs are sufficiently large, it would be sensible to use a scheme that mixes much better for the same computational cost. If the aim is as in 2(b) then there will be much more interest in the impact of using Milstein within a framework that is currently being used by statisticians.

6. Geometric Brownian motion is not a “typical” example. Because the drift and volatility coefficient are both linear, second derivatives are zero and so some of the terms neglected by both the Euler and Milstein schemes are in fact zero. Further, it does not have a proper stationary distribution, leading to the growth visible in the plots. I would recommend switching the places of the CIR and the GBM in the document. The CIR is more typical of many of the SDEs on which people actually perform inference.

Medium points

- Figure 1. This provides evidence for the issue suggested in point 4 above. There is little to distinguish the schemes, so perhaps you have not found SDEs, or discretisation levels of the SDEs where Milstein really leads to a big improvement over Euler. Since it requires more effort, the gains are, therefore, not worth the extra computational cost.
- After first paragraph of Section 2.1: given that this is the difference between the schemes, it would be worth stating what the order of strong convergence is (*i.e.*, define it) and explain why it is important. (One paragraph).

3. It would be worth making the point, from P7L31-33, directly, that for *any* 1-D diffusion, because of the quadratic term in B_k there must always be an upper (if multiplier is negative) or lower bound (if positive) on the size of the increment. Then you can, on P9, point out the consistency with the bounds that come out of the likelihood. Or, maybe better, derive the bounds from the scheme itself, which is simpler, and then connect these to the likelihood.
4. P10-11. You are doing individual random walks or multiplicative random walks on the parameters. Typically some parameters are highly correlated and single updates can be inefficient. Much better (and standard these days) to update the parameters (e.g. $(\theta_1, \log \theta_2, \dots)$, if $\theta_1 \in \mathbb{R}$ and $\theta_2 \in \mathbb{R}^+$) as a block using a random walk, using a short tuning run to estimate a sensible variance matrix. If you wish to stick with the current approach then it would be worth providing a plot in the appendix to show that the two parameters are not strongly correlated.
5. Wrt major point 5: P15L28 “All estimation methods compared here are affected by this issue in the same way; we hence do not further consider it here.” Estimation methods have move on a great deal in the last 18 years. For a Metropolis-within-Gibbs scheme as you use, then then the scheme in [6] represents the state of the art and would be appropriate. However, most researchers now use particle MCMC (with an appropriate bridge such as the MB that is used here), mainly for its simplicity: here you could still use Euler or Milstein for the simulation and the likelihood, and the results you find would be more relevant to practitioners today.
6. Section 4. I had not appreciated that the implementation of the MB for the Milstein scheme would be so tricky and involve numerical optimisation (for robustness) and then integration. An alternative to the the scheme that is proposed seems to be appropriate, based on an approximation. The form for π_{Euler} is highly suggestive of a drift, $(X_{\tau_{i+1}} - X_{t_k}^*)/(\tau_{i+1} - t_k)$ and volatility $(\tau_{i+1} - t_{k+1})/(\tau_{i+1} - t_k)\Sigma(X_{t_k}^*; \theta)$ for the bridge process; i.e. the diffusion *conditioned* on $X_{\tau_{i+1}}$. So, why not use the Milstein formula directly arising from this approximation to the conditioned process? The MB can be viewed as doing something similar - obtains an (the same) approximate drift and volatility for the conditioned process and then applies the Euler density with this. With perfect numerics, it will not be as accurate as what is attempted in the paper, but it will certainly be much quicker, and since numerics are imperfect, it may well end up being more accurate.

Minor points

- P5 L46 one of the ΔB_k s still has a (j) superscript.
- P8 L43 random walk is more specific than this. e.g. MALA would satisfy the definition given in the paper. The proposal has to be additively symmetric about θ to be a random walk - and it *can* depend on the X values and still be a random walk on θ .
- Reference [10] is in an odd format, with 3 author names and then ‘et al.’
- Reference [11] ‘heston’ should have a capital H.

Appendix B

Response to Comments from the Associate Editor and Reviewers

June 8, 2020

We thank the Associate Editor and the reviewers for their helpful comments and for the invitation to revise our manuscript, which we have carefully done. In the following, we provide point-by-point answers, with received comments in black and our replies in blue. Moreover, we attach a version of the updated manuscript where the changes are also marked.

Comments by Associate Editor

We have two good reports from the referees. They have both raised a number of points which need addressing before we can move to acceptance. There is quite a lot to address, but even if no action is needed after meditation on some of the points, please prepare a careful response.

Comments by Reviewer 1

Summary

Inference on the parameters, and potentially the paths, of SDEs is typically performed by approximating the paths through a discretisation and then using the Euler-Maruyama method to estimate the likelihood. Typically the Euler-Maruyama (E-M) method is also used to create the proposed paths, whether through forward simulations or some kind of conditioning on future observations. The main exception to this is in those rare cases (such as in one dimension) where the Exact Algorithm or its evolutions can be used to perform exact inference. This article examines replacing the E-M method by the Milstein method, which contains an additional ‘correction’ term and has a higher order of accuracy. The use of the method is described, including the myriad difficulties and problems, and then the method is tested on the 1D example of Geometric Brownian Motion (GBM) [and the CIR process in the appendix]. Little if any improvement in accuracy is found, and the computational cost is much higher than for the E-M method.

Main points

This has the potential to be an interesting paper. To my knowledge, no-one else has looked at the question of replacing E-M with Milstein, yet it is a natural question. This paper starts to answer it, and describes the fundamental issues well, but needs much clearer and better thought-out simulation studies.

0. I certainly had not appreciated the difficulties with using the Milstein scheme in dimension > 1 , and this is clearly pointed out in the paper. Further, even in one dimension, the Milstein scheme has an odd property that because it is quadratic in ΔB the increment has either an upper or lower bound, and this follows through to issues with the likelihood. This is discussed in the paper, though it might be a clearer connection can be made between the scheme itself and the likelihood issues (see medium point 3).

We thank the reviewer for making this point. We have made some modifications of the text, described in the reply to medium point no. 3 below.

It also seems to be that the set of cases where the Milstein scheme can be used is only slightly larger than the set for which the EA is valid, so at least a brief discussion of why one might prefer Milstein seems warranted.

Using Lamperti’s transform, any time-homogeneous one-dimensional diffusion process can be transformed to a diffusion process with unit diffusion, to which the EA1 applies. For multidimensional diffusions, the MCMC scheme with Milstein approximation would require the stochastic double integral in the last term of Equation (7) in our manuscript to yield an analytical solution in order to avoid another layer of approximation. Sufficient conditions are listed on page 5. The EA2 and EA3, which are applicable to multidimensional diffusions, formulate requirements via the Radon-Nikodym derivative of the target and the proposal measure. Hence, for one-dimensional diffusions, the Exact Algorithm is an alternative to the MCMC scheme considered here, regardless of whether transition densities are approximated by the Euler or the Milstein scheme. For multi-dimensional diffusions, one has to check which of the methods is applicable.

The aim of our manuscript is to study whether the computational performance of the generic and easily applicable MCMC scheme can be improved by using higher-order approximations of the transition density of the diffusion process. The limited applicability of the scheme in case of the Milstein approximation is a result from our study. We have added a sentence to the discussion.

1. I am missing the motivation to use the Milstein scheme for proposing a path and the Euler scheme for evaluating the likelihood, or vice versa. Why would this be a consistent, sensible approach?

Also, it would be good to have confirmation that if the Milstein scheme is used to propose a path then the q for that path, needed in the acceptance ratio, is also calculated using the Milstein scheme; this has to happen for consistency, and also surely would potentially lead to issues when simulations are by Euler yet likelihood by Milstein that certain proposals would have a proposal probability of zero because of point 0 above.

Yes, the q for the path used in the acceptance probability is always based on the same scheme as was used to propose the path. We are not sure which part of the manuscript might have indicated otherwise. The only part that we found that might have been ambiguous was in Section 4 about the implementation where we now write:

”For the combination of the MB proposal and the Milstein approximation, the set of feasible proposal points may be empty. In this case, our implementation shifts to the Euler approximation for this point, i.e. the point is proposed with the MB proposal based on the Euler scheme and also the corresponding factor of the proposal density in the acceptance probability is based on the Euler scheme.”

Unless there is a good justification that I’m missing, I would suggest just two scenarios here, rather than four: Milstein for everything or E-M for everything.

We had considered the ”Euler-Milstein combination”, i.e. an Euler proposal combined with a Milstein likelihood approximation, for the following reason: The Euler scheme is faster than Milstein for proposing a path, and since the proposal is just a proposal, the accuracy might be of less importance. For the likelihood evaluation as part of computing the acceptance probability, however, the accuracy might be more crucial, and hence we used the Milstein scheme here. The other mixed combination, ”Milstein-Euler”, was included mainly for the sake of completeness. We agree that it may not be reasonable. Therefore, we dropped this combination in the new version of the manuscript.

2. There are two factors being confounded. (a) The accuracy of the approximation to the posterior distribution given a particular discretisation (m) and SDE integration scheme (Euler/Milstein), and (b) the accuracy with which that approximate posterior distribution has been obtained by the MCMC; the mixing of the MCMC depends on the choice of bridge scheme: left-conditioned (i.e., no bridge) or MB. Also, with the MCMC scheme used, the mixing degrades with m . Interest might be purely in (a) but you need to control for (b) or you might be interested in the overall best accuracy obtainable given a fixed amount of computer time. I think both would be of interest, but from the way the results are presented it seems like only the latter is considered, and even then, this is not made explicit. To consider the former you would need to ensure that all schemes are run for long enough so that that you are confident any differences in inference (e.g. posterior means) arise from (a). This is straightforward to ensure given e.g. the ESSs and the posterior variances.

We had opted for a fixed number of iterations instead of a fixed amount of computational time mainly in order to ensure reproducibility. However, we understand the reviewer’s concerns and therefore, reran the simulation study and completely revised the corresponding sections (5. Simulation study, 6. Results, 7. Discussion). We now let each procedure run for a fixed amount of computational time of one hour and evaluate the overall accuracy.

3. The wrong comparisons are being made. Currently 100 data sets are simulated and the distribution of posterior means under the different schemes are compared with the distribution of the true MAP estimates and the MLEs. This (a) *is not comparing like with like* and it (b) *is not using the fact that the same 100 data sets are used*. For (a) since GBM is tractable, you can obtain the true posterior distribution for the parameters (as you must have used to obtain the MAP), so run a *really long* MCMC on the two parameters and find, to very high accuracy, the *true* posterior mean for each data set. For (b) you then pair the results from a particular approximate run on a particular data set with the truth for that same data set. You can then provide (i) density plots of these *discrepancies* (i.e. approximating the distribution, over data sets, of the error in the estimate of the posterior mean), and (ii) RMSEs for these quantities. Ideally, if your interest is in ”accuracy after a given computational time” you would use the same computational time for all runs. Currently it is entirely unclear what Tables 1 and 2 depict; I would guess that what is reported as ”bias” is actually the deviation from the true parameter value; similarly RMSE. Or perhaps they are looking at deviations from the MAP values? But this is not what you want to compare against, as already stated. The posterior for σ^2 , at least, is not symmetric so the MAP and posterior mean are not the same. I would also be tempted to look at posterior medians rather than (or in addition to) posterior means, since they are more robust. Ideally you would also look at estimates of the posterior variance.

We have taken this comment very seriously and revised the simulations and result presentations. For the new version of the manuscript, we proceed as proposed: We calculated and compared the posterior mean, median, and variance of the approximated posterior distribution based on the different approximation methods and sampling schemes and the posterior mean, median, and variance of the true posterior distribution. We present the density plots of the discrepancies and the RSMEs calculated for the discrepancy of the three quantities.

4. The simulation study comparisons may have been doomed from the start. It may be that the inter-observation time is so small that there is very little difference between. E-M and Milstein as both are very accurate - and far more accurate than the MCMC is able to discern given the 10^5 iterations. One would expect to see this e.g. in Table 1 for $m = 2$, where Milstein-Milstein should have a lower bias and RMSE than E-M-E-M. It does not. This may be down to the point being made here, or that made in 3. or both. At the moment I cannot tell.

We believe that the lack in differences was due to the point made in comment 3. We have carefully revised the simulation study and now do see differences in the results for the considered methods.

- 4.b) The left-conditioned scheme usually performs terribly when observations are exact (at least for $m \geq 5$). The fact that it does as well as the MB here suggests indeed that the observations are too close together to allow much freedom. Given that (as is well known and demonstrated in Figure 4) the left-conditioned scheme is usually poor in the case of exact observations, and given that these schemes only affect the efficiency of the MCMC, not the accuracy of the approximate posterior if the MCMC were carried on to ∞ , it would not be unreasonable to drop it the left-conditioned scheme. One alternative possibility would be to replace this bifurcation with two uses of the MB: as in [6] and within particle MCMC.

We followed the reviewer’s advice and dropped the left-conditioned scheme from the simulation study. For a better overview of the remaining results, we restrict the study to this scheme. We exclude the innovation scheme [6] from the simulation study for reasons explained in the answer to comment 5. For the sake of clarity and comprehensibility, we kept the description of the left-conditioned scheme in Section 3.2.

5. The MCMC scheme being used was state-of-the art 20 years ago. The mixing criticism is acknowledged in the article, but still, inference has moved on considerably since then. In terms of latent variable schemes similar to the one used, that in [6, Golightly and Wilkinson, CSDA] is hard to beat; however, nowadays the usual method is particle MCMC. I bring this point up for two reasons. If the aim is purely the one in 2(a) then you want the MCMC to mix as thoroughly as possible as quickly as possible, so unless it is feasible to increase the number of iterations of your current scheme until all the ESSs are sufficiently large, it would be sensible to use a scheme that mixes much better for the same computational cost. If the aim is as in 2(b) then there will be much more interest in the impact of using Milstein within a framework that is currently being used by statisticians.

The original aim of the study was to assess the overall accuracy (2a and 2b) of the MCMC scheme as described in the manuscript and, building on that, as in [6, Golightly and Wilkinson, CSDA]. We started with the standard approach and met with obstacles as described. These will persist in the innovation scheme: The scheme leaves the approximation of the likelihood unaltered and starts off with an equivalent proposal sampling; the difference lies in a deterministic transformation of the imputed path once a new parameter is accepted and vice versa. Hence, the limited applicability to multivariate diffusions and the bounds from Equations (10) and (11) will persist, which we have now briefly added to the discussion. For that reason, we did not consider it promising to analyze the Milstein approximation also for the innovation scheme. The results of our study are rather negative. However, we would have been glad if we had found such investigations described anywhere; as the reviewer says, the question is a natural one. In the spirit of open research, we believe in the added value of publishing the insights gained.

6. Geometric Brownian motion is not a “typical” example. Because the drift and volatility coefficient are both linear, second derivatives are zero and so some of the terms neglected by both the Euler and Milstein schemes are in fact zero. Further, it does not have a proper stationary distribution, leading to the growth visible in the plots. I would recommend switching the places of the CIR and the GBM in the document. The CIR is more typical of many of the SDEs on which people actually perform inference.

We tried to make the manuscript readable also for readers of Royal Society Open Science that are not that familiar with the topic and stated this in the manuscript (“In this work, we use a simple, well-known example of such a diffusion process in order to focus on the investigated estimation methods and make the article easy to follow.”). The property of having a stationary distribution is not a prerequisite to perform inference, especially because one cannot assume that an observed process has reached its stationary distribution during the time of observation. Also, while the transition density of the CIR process involves the sum of an infinite series, the GBM has a simple explicit transition density, the density of the log-normal distribution. Thus, we consider it more suitable as an illustrative example. Moreover, we provide all the relevant details for the CIR process in the appendix. Therefore, we prefer to leave the order of examples as it is.

Medium points

1. Figure 1. This provides evidence for the issue suggested in point 4 above. There is little to distinguish the schemes, so perhaps you have not found SDEs, or discretisation levels of the SDEs where Milstein really leads to a big improvement over Euler. Since it requires more effort, the gains are, therefore, not worth the extra computational cost.

In the revised simulation study, where we especially followed the suggestions from main point 3 above, differences become apparent in the results for the different methods, which is in line with main point 4.

2. After first paragraph of Section 2.1: given that this is the difference between the schemes, it would be worth stating what the order of strong convergence is (i.e., define it) and explain why it is important. (One paragraph).

We added the following paragraph:

"A discrete-time approximation \mathbf{Y} with maximum step size $\Delta > 0$ converges with strong order $\gamma > 0$ at time T to the solution X_T of a given SDE, if there exists a positive constant C independent of Δ and a $\Delta_0 > 0$, such that

$$E(|X_T - \mathbf{Y}_T^\Delta|) \leq C\Delta^\gamma,$$

for all $\Delta \in (0, \Delta_0)$. Strong convergence ensures a pathwise approximation of the solution process $(X_t)_{t \geq 0}$ of the given SDE. The higher the order of strong convergence is, the faster the mean absolute error between the approximation and the solution decreases as the maximum time step size Δ decreases."

3. It would be worth making the point, from P7L31-33, directly, that for *any* 1-D diffusion, because of the quadratic term in B_k there must always be an upper (if multiplier is negative) or lower bound (if positive) on the size of the increment. Then you can, on P9, point out the consistency with the bounds that come out of the likelihood. Or, maybe better, derive the bounds from the scheme itself, which is simpler, and then connect these to the likelihood.

We added the following paragraph at the end of Section 2.1. and referred to it in Section 2.2.

"Moreover, note that if we consider the approximation $Y_{k+1}^{(i)}$ in Equation (8) as a function $g(\Delta B_k^{(j)})$ of the increment of the Brownian motion, g is quadratic in $\Delta B_k^{(j)}$. Therefore, the function g has a global extremum with value

$$g^* = Y_k^{(i)} - \frac{1}{2}\sigma_{ij}(Y_k, \theta) \left(\frac{\partial \sigma_{ij}}{\partial y^{(i)}}(Y_k, \theta) \right) + \left(\mu_i(Y_k, \theta) - \frac{1}{2}\sigma_{ij}(Y_k, \theta) \frac{\partial \sigma_{ij}}{\partial y^{(i)}}(Y_k, \theta) \right) \Delta t_k.$$

Hence, there is a bound on the range of possible values for $Y_{k+1}^{(i)}$ resulting from the Milstein scheme which might exclude values that the solution process X_{t_k} could take. Whether this is a lower or upper bound depends on the sign of the diffusion function and its derivative. The second derivative of g is given by

$$\frac{\partial^2 g(\Delta B_k^{(j)})}{\partial (\Delta B_k^{(j)})^2} = \sigma_{ij}(Y_k, \theta) \frac{\partial \sigma_{ij}}{\partial y^{(i)}}(Y_k, \theta) =: g''.$$

Thus, the extremum g^* is a maximum and puts an upper bound on the possible values of $Y_{k+1}^{(i)}$ if $g'' < 0$, and g^* is a minimum and puts a lower bound on $Y_{k+1}^{(i)}$ if $g'' > 0$. For the case where $g'' = 0$, the Milstein scheme reduces to the Euler scheme."

4. P10-11. You are doing individual random walks or multiplicative random walks on the parameters. Typically some parameters are highly correlated and single updates can be inefficient. Much better (and standard these days) to update the parameters (e.g. $(\theta_1, \log \theta_2, \dots)$, if $\theta_1 \in \mathbb{R}$ and $\theta_2 \in \mathbb{R}^+$) as a block using a random walk, using a short tuning run to estimate a sensible variance matrix. If you wish to stick with the current approach then it would be worth providing a plot in the appendix to show that the two parameters are not strongly correlated.

In the new version of the manuscript, we provide plots showing that the parameters are not strongly correlated in Appendix E.

5. Wrt major point 5: P15L28 "All estimation methods compared here are affected by this issue in the same way; we hence do not further consider it here." Estimation methods have move on a great deal in the last 18 years. For a Metropolis-within-Gibbs scheme as you use, then then the scheme in [6] represents the state of the art and would be appropriate. However, most researchers now use particle MCMC (with an appropriate bridge such as the MB that is used here), mainly for its simplicity: here you could still use Euler or Milstein for the simulation and the likelihood, and the results you find would be more relevant to practitioners today.

We are aware of the advances in the field. However, as described in the reply to major point 5, the limitations of the Milstein approximation would still persist in the innovation scheme and particle MCMC. We have added this to the discussion. We hence prefer to focus the simulation study on the basic MCMC scheme.

6. Section 4. I had not appreciated that the implementation of the MB for the Milstein scheme would be so tricky and involve numerical optimisation (for robustness) and then integration. An alternative to the the scheme that is proposed seems to be appropriate, based on an approximation. The form for π_{Euler} is highly suggestive of a drift, $(X_{\tau_{i+1}} - X_{t_k}^*)/(\tau_{i+1} - t_k)$ and volatility $(\tau_{i+1} - t_{k+1})/(\tau_{i+1} - t_k)\Sigma(X_{t_k}^*; \theta)$ for the bridge process; i.e. the diffusion *conditioned* on $X_{\tau_{i+1}}$. So, why not use the Milstein formula directly arising from this approximation to the conditioned process? The MB can be viewed as doing something similar - obtains an (the same) approximate drift and volatility for the conditioned process and then applies the Euler density with this. With perfect numerics, it will not be as accurate as what is attempted in the paper, but it will certainly be much quicker, and since numerics are imperfect, it may well end up being more accurate.

We would like to thank the reviewer particularly for this suggestion. We have added the proposed strategy to Section 3.2. under the name *diffusion bridge Milstein*, and combined with the Milstein scheme for the approximation of the likelihood (DBM-M), also included it in the simulation study.

This combination turned out to be the most effective in terms of overall accuracy for estimates of the diffusion parameter.

Minor points

- P5 L46 one of the ΔB_k s still has a ^(j) superscript.
was corrected

- P8 L43 random walk is more specific than this. e.g. MALA would satisfy the definition given in the paper. The proposal has to be additively symmetric about θ to be a random walk - and it can depend on the X values and still be a random walk on θ .

We agree that our original description of the random walk proposal was not very precise. However, random walk proposals are not necessarily symmetric by definition (see e.g. Tierney, 1994). We rephrased the description at the beginning of Section 3.1. as follows:

”If a proposal $\theta^* = \theta + u$ with an update u that is independent of the current parameter value θ is used, the proposal strategy is called a random walk proposal.”

- Reference [10] is in an odd format, with 3 author names and then ‘et al.’
was corrected
- Reference [11] ‘heston’ should have a capital H.
was corrected

Comments by Reviewer 2

The paper compares the Euler-Maruyama and Milstein approximations to the transition density of a stochastic differential equation. The authors compare the approximations using two data augmentation Markov chain Monte Carlo algorithms. The first uses the (approximate) transition density to propose new points and the second uses the modified diffusion bridge. An alternate derivation of the modified diffusion bridge using the Milstein approximation is provided. The authors test all combinations of the two approximations (both for parameter updates and proposals) on the two data augmentation approaches, which leads to 8 different algorithms. They apply these algorithms on a geometric Brownian motion SDE as well as a Cox-Ingersoll-Ross example in Appendix D. The paper was interesting to read and the results were quite surprising. However, there are a few issues that should be addressed.

Main comments

1. As I understand it, the lower bound for the Milstein scheme disallows feasible values of X . In the Milstein MB proposal, no values below the lower bound are sampled and when the Milstein approximation to the transition density is used, values below the lower bound are given 0 weight. Is this correct?

Yes, values beyond the bound are given zero weight. Whether the bound is an upper or a lower bound depends on the sign of the diffusion function and of its derivative for the specific example.

2. How is the lower bound affected by the level of discretisation? A comparison of the exact transition density and the Euler-Maruyama and Milstein approximations is shown in Figure 2. In Figure 2b, the error in the Euler-Maruyama approximation can be made arbitrarily small by decreasing the time step. Can the same be said for the Milstein approximation in that example (in the context of the lower bound)?

For the value of the bound derived in Section 2.2., we have

$$Y_k - \frac{1}{2} \frac{\sigma_k}{\sigma'_k} + \left(\mu_k - \frac{1}{2} \sigma_k \sigma'_k \right) \Delta t_k \longrightarrow Y_k - \frac{1}{2} \frac{\sigma_k}{\sigma'_k}, \text{ as } \Delta t_k \rightarrow 0.$$

The transition density $\pi(X_{t_{k+1}} | X_{t_k} = Y_k)$ of the solution X conditioned on $X_{t_k} = Y_k$ increasingly concentrates in a narrow interval around Y_k as Δt_k decreases. This follows from the fact that diffusion processes have almost surely continuous sample paths. Thus, the weight of the distribution of $X_{t_{k+1}} | X_{t_k} = Y_k$ that falls beyond the bound derived from the Milstein scheme tends to 0 as the time step Δt_k tends to 0.

Moreover, as pointed out in Section 2.1. (more elaborately in the new version of the manuscript), *both* approximations are strongly converging, i.e. the mean absolute error between the approximation and the solution of a given SDE converges to 0 as the time step tends to 0.

3. If the lower bound cuts off feasible values, won't this bias the results? In practice, how does the lower bound affect inference results? In the simulation study in section 5, are the marginal posteriors for θ the same for all 8 methods?

Any approximation will bias the results. As pointed out in the reply to the previous comment, the lower bound is not problematic for small time steps between observed/imputed points. For the method that combines the modified-bridge proposal with the Milstein proposal, larger time steps might occur because it involves the transition density for transitioning from the point that we would like to propose next ($X_{t_{k+1}}$) to the next observed point to the right (X_{t_m}). Moreover, the proposal density of this method consists of the product of two transition densities based on the Milstein scheme which may result in an upper and a lower bound. If the upper bound is smaller than the lower bound, the set of feasible points for the modified-bridge Milstein proposal would be empty and we would be forced to switch to a different proposal method. In our simulation study, this case did not occur.

The marginal posteriors for θ are not all the same in the simulation study. This becomes more apparent in the new version of the manuscript.

4. Is there some intuition about why the Milstein approximation does not perform better than the Euler-Maruyama in the simulation study?

We have revised our simulation study and now do see differences in the performance of the two approximation schemes. The overall accuracy achieved by the methods that use the Milstein approximation is consistently higher for estimates for the diffusion parameter which seems intuitive since the additional term added by the Milstein scheme compared to the Euler scheme involves the diffusion function and its derivative.

5. In the GBM example, the Euler-Maruyama and Milstein approximations give very similar results. In general, the Milstein scheme yields a more accurate approximation to the transition density than the Euler-Maruyama approximation. However, if the latter is already a good approximation, then the Milstein scheme would not necessarily improve the results.

We are aware of this issue. However, neither the choice of the example model (GBM) nor the specific parameter combination chosen were the reason why we could not see any differences in the performance of the two approximations, but it was rather due to the way we had designed the simulation study and analysed the results. For the new version of the manuscript, we have carefully revised both sections

and are now able to point out differences in the performance of the two approximations based on the same model and parameter combination as before.

6. Related to the above point. In Figure 2b, the Euler-Maruyama approximation with a time step of 0.1 is not very accurate, but in the simulation study it yields quite good results. The time step used in the simulation study (with no imputation) is 0.02 however. How accurate is the Euler-Maruyama approximation in this case? Please give a comparison/measure of the accuracy of the two approximations for the simulation study in section 5.

In the new version of the manuscript, we use the root mean square error (RMSE) to evaluate the overall accuracy (i.e. the combination of how well the true posterior distribution is approximated and how well we can sample from the approximated posterior distribution) of the different estimation methods. For each of the 100 simulated datasets, we let each of the considered methods run for one hour to obtain a sample from the corresponding approximated posterior distribution. Moreover, we draw a large sample from the true posterior distribution based on the solution of the benchmark model. For both posterior samples, we calculate several statistics (mean, median, variance). We take the difference between the respective summary statistic of the sample from the approximated posterior and the sample from the true posterior and calculate the RMSE aggregated over these 100 difference values, one for each dataset.

7. In the simulation study, 50 equidistant points are taken from each path. Are the results of the comparison the same when less points are used (e.g. 10 or 20)?

We have run the simulation study for 10, 20, and 50 equidistant observed points. The results of the comparison are qualitatively the same; therefore, we only present one of the scenarios ($M = 20$) in the manuscript. The results for the other scenarios will be available in the github repository mentioned in the manuscript.

Minor comments

- In Section 2.1, the stochastic Taylor expansion of the process is not defined.

We omitted the general definition of the stochastic Taylor expansion because one would usually only write out the terms that are included in the Milstein approximation anyway plus a "complicated" remainder term. We believe that adding this formula would not contribute to the legibility of the article, and instead, refer the interested reader to Kloeden, Platen (1992) for a general treatment of the stochastic Taylor expansion.

- Typo in the first equation in Section 2.2 was corrected

Appendix C

Response to Comments from the Associate Editor and Reviewers

September 14, 2020

We thank the Associate Editor and the reviewers for their helpful comments. In the following, we provide point-by-point answers, with received comments in black and our replies in blue. Moreover, we attach a version of the updated manuscript where the changes are also marked.

Associate Editor Comments to Author (Professor Andreas Kyprianou)

There are still quite a few things that the referees have highlighted but they are all willing to allow the corrections in good faith and proceed to accept with minor revisions. I am recommending accordingly.

Comments by Reviewer 1

This version of the manuscript really is a substantial improvement. The simulation study, in particular, is now much clearer and more informative. I have two "medium" comments and several minor comments.

1. In the simulation study it would still be good to be clear between 2(a) and 2(b), Specifically, how much are the MSEs of the posterior quantities down to approximation error and how much down to not a high enough ESS. I suspect it's nearly all down to approximation error - but can you quantify this? e.g. for the MSE for the mean, you also have a typical size of the posterior variance, you know the ESS, so the contribution to the MSE from MCMC variance can be calculated. The remainder must be bias². Is one much larger than the other?

Unfortunately, we were not completely sure whether we have understood this comment correctly; therefore, we have not made any changes to the manuscript regarding this comment, but provide what we suspect was requested here.

The following tables contain the MSE of the mean. These are equal to the square of the RMSE of the mean as in Tables 1 and D1 (columns 1 and 4) in the manuscript. Moreover, we calculated the median \overline{Var} of the estimated posterior variance for the samples from the approximated posterior distributions assuming this was meant by the "typical size of the posterior variance". We determined the effective sample size \overline{ESS} for the corresponding samples from the approximated posterior distributions and calculated the square root of the ratio of these two quantities as an attempt to quantify the Monte Carlo error.

GBM

Method		Results for α		Results for σ^2	
		MSE of mean	$\sqrt{\overline{Var}/\overline{ESS}}$	MSE of mean	$\sqrt{\overline{Var}/\overline{ESS}}$
$m = 1$	Euler	0.0793	0.0019	0.4072	0.0003
	Milstein	0.7250	0.0036	0.0796	0.0017
$m = 2$	MBE-E	0.0710	0.0034	0.0447	0.0011
	MBE-M	0.0967	0.0080	0.0119	0.0037
	MBM-M	0.0989	0.0196	0.0125	0.0066
	DBM-M	0.1012	0.0080	0.0102	0.0033
$m = 5$	MBE-E	0.0766	0.0045	0.0128	0.0029
	MBE-M	0.0830	0.0127	0.0010	0.0097
	MBM-M	0.0853	0.0451	0.0016	0.0265
	DBM-M	0.0845	0.0129	0.0010	0.0095

MSE denotes the mean square error.

CIR process

Method		Results for α		Results for σ^2	
		MSE of mean	$\sqrt{\text{Var}/\text{ESS}}$	MSE of mean	$\sqrt{\text{Var}/\text{ESS}}$
$m = 1$	Euler	0.0003	0.0007	0.0257	0.0003
	Milstein	0.0003	0.0016	0.0171	0.0006
$m = 2$	MBE-E	0.0001	0.0013	0.0083	0.0009
	MBE-M	0.0001	0.0027	0.0043	0.0019
	MBM-M	0.0002	0.0086	0.0043	0.0060
	DBM-M	0.0001	0.0028	0.0043	0.0019
$m = 5$	MBE-E	0.0000	0.0017	0.0016	0.0026
	MBE-M	0.0001	0.0045	0.0007	0.0071
	MBM-M	0.0009	0.0233	0.0026	0.0370
	DBM-M	0.0001	0.0047	0.0007	0.0066

2. You have different acceptance rates for the different m values, and are reporting this. Usually one tunes a RWM (of log RWM) algorithm to obtain a particular acceptance rate (around 40% in dimension 1, I think). The different acceptance rates are actually suggesting that the tunings may not be comparable - different scalings could have been better for different m values. This needs mentioning. However, I'm not suggesting you redo the simulation study; instead you could, for example, refer to the Efficiency vs acceptance rate curve in Fig 3 of Roberts and Rosenthal (2001), Stat. Sci., which shows that (at least in the high-dimensional limit) efficiency is pretty robust to changes in the acceptance rate of around 0.1 either side of the optimum. So your ESS's are probably comparable to within a factor of two, or even tighter.

We added the following comment to the discussion:

“Moreover, note that tuning the variance hyperparameters for the random walk proposals of the parameters in Steps 3a and 3b in the simulation study to reach an optimal acceptance rate might lead to a higher ESS. However, since the acceptance rates achieved in the simulation study lie in a range where the sampling efficiency is rather robust to changes in the acceptance rate as shown in Roberts and Rosenthal (2001) (in the high-dimensional limit), we do not expect the change in the ESS after tuning to be substantial.”

MINOR

- P17 the first paragraph talks about a sample from the true posterior, but it's not until the third paragraph that this is described. Perhaps add "(see below)" or similar to the first paragraph?
was added

- p17 "obtain a two-dimensional Markov chain". Pedantic, but strictly taking (α, σ^2) from all your MCMC schemes that also involve imputed X values, the set of these two parameter values is not Markovian. Please use a different description.

We rephrased the sentence as follows:

“From each estimation procedure, we obtained an MCMC chain of dimension $n(m - 1) + 2$. For each chain, we used the two components for parameters α and σ^2 and calculated the mean, the median, and the variance after cutting off a burn-in phase of 5000 iterations.”

- P18 100 of such → a hundred such
was corrected
- P19 data is a plural word, so "when data are imputed" and "data are introduced"
was corrected
- P21 "This strength [of the Euler scheme] is weakened by the limited applicability of the Milstein approximation..." I cannot make sense of this statement.

We rephrased the sentence as follows:

“This strength does not translate to the Milstein-based MCMC scheme due to the limited applicability of the Milstein approximation especially in the multidimensional setting.”

- P21 Less iterations → Fewer iterations
was corrected
- P22 It was expected does make → It was expected ... would make
was corrected

Comments by Reviewer 2

The authors have done a great job with the revision. I only have one comment regarding the DBM proposal function.

1. In the simulation studies, the DBM-M and MBE-E methods give the best overall accuracy for the diffusion and drift coefficients, respectively. In the latter case, the computational efficiency of the Euler scheme outweighs any increase in accuracy provided by the Milstein scheme when approximating the transition density. Moreover, the DBM-M combination generally seems to outperform the MBE-M combination in terms of overall accuracy and multivariate ESS. Considering this, I believe the combination DBM-E would also be of interest. The results from this combination (especially when compared to MBE-E) would give an indication of the usefulness of the Milstein scheme (or other higher-order approximations) for proposal functions. Even though DBM is not as generally applicable as MBE, it might be an interesting area of future research.

We had considered several "mixed" combinations, i.e. estimation procedures that combine the Euler and the Milstein scheme, in the previous version of the manuscript. In the revised simulation study, we only include Combination MBE-M because it combines the faster scheme for the proposals (where accuracy is less important) and the more accurate scheme for the acceptance probability. Reverse combinations (that use the less accurate scheme for the acceptance probability) do not seem reasonable. This was pointed out to us by Reviewer 1 and we agreed.

2. Typo in Equation (13): τ_1 is used in the product instead of τ_i
was corrected

Comments by Reviewer 3

The paper treats a relevant question - in fact, I am slightly surprised that this has not been addressed before in the literature although the reason may be that the answer is largely negative: few additional gains are to be made from using the Milstein scheme over an Euler scheme in Bayesian parameter estimation for diffusion processes.

Other than some minor comments (mostly typographical or stylistic but also one point where I believe the paper is mathematically incorrect), I feel that the paper, now in its second round, is essentially ready for publication.

- p3 l38: Maybe it would be good to point to the CIR example in appendix here already. I accept the authors' argument about not reversing the position of CIR vs GBM in appendix vs paper in the interest of having as accessible an example as possible in the paper. Nonetheless, the previous criticism that many higher order terms in the stochastic Taylor expansion are zero in the GBM case stands and it is therefore important to make clear that the content applies to other SDEs where this is not the case, too.

We rephrased the paragraph as follows:

“In this work, we use rather simple, well-known examples of such a diffusion process in order to focus on the investigated estimation methods and make the article easy to follow. Our example for the main text is the geometric Brownian motion (GBM), and in Appendix D, we also provide all relevant details for the Cox-Ingersoll-Ross (CIR) process. The GBM is described by the following SDE: ...”

- p4 l 43 ”stochastic Taylor expansion of the process $(X_t)_{t \geq 0}$, and has strong order of convergence 0.5.” would be my suggested phrasing
was corrected

- p4 l45 The notation \mathbf{Y} is a bit problematic: while fairly intuitive, it seems to be neither defined nor used at all elsewhere in the paper. Also, it once arises with the superscript Δ and once without.

We removed the boldface font and include the superscript Δ for every occurrence of Y .

- p5 l38 claims that the equality at line 35 following ”More generally, we require that” (eliminating those stochastic double integrals that cannot be computed analytically from the Milstein approximation) holds if and only if sigma has at most one non-zero entry per row. I believe that this condition is sufficient but not necessary. I believe the undesirable double stochastic integrals also disappear if sigma is constant but full. Another counter example is the matrix $(2+\cos(x),0;1,1)$ (first row with entries $2+\cos(x)$ on diagonal and zero off diagonal, second row has both entries equal to one). This makes the Milstein scheme slightly more widely applicable than claimed by the authors but overall the conditions are still rather restrictive, so I don't think much else needs to change in the paper.

We rephrased the whole paragraph as follows:

“More generally, we require that

$$\sigma_{rj}(Y_k, \theta) \frac{\partial \sigma_{il}}{\partial y^{(r)}}(Y_k, \theta) \equiv 0 \quad \text{for } j \neq l \quad (1)$$

so that only $j = l$ is inside the double integral. Relation (1) implies the following:

- if an entry $\sigma_{rj}(Y_k, \theta)$ is non-zero, then the entries of all *other* columns and *all* rows must not depend on $Y_k^{(r)}$, and
- if an entry $\sigma_{il}(Y_k, \theta)$ depends on $Y_k^{(r)}$, then the entries of all *other* columns in row r must be zero.

In particular, this means that unless the r^{th} row of the diffusion function contains only zeros, component $Y_k^{(r)}$ can only appear in *one* column of the diffusion function (and if it appears, then the entries of all *other* columns in row r must be zero). Moreover, each component of the diffusion process $(X_t)_{t \geq 0}$ can only be directly affected by more than one component of the Brownian motion, if the size of all stochastic effects (i. e. *all* entries of the diffusion function) does not depend on the respective component of the diffusion process. Further, if all d components of the diffusion process appear in the diffusion function, then the process can be affected by at most d components of the Brownian motion. Besides, if all d components of the diffusion process appear in the diffusion function and the process shall be affect by d components of the Brownian motion, the diffusion function must be a (possibly column-wise permuted) diagonal matrix. In many applications, these are not realistic assumptions.

Assume that the i^{th} component of the diffusion process appears in the i^{th} row of the diffusion function and that the respective entry of the diffusion function does not depend on the remaining components $Y_k^{(r)}$, $r \neq i$ (the contrary would impose restrictions on other rows, as described above). Then, the i^{th} component of the approximated process is

$$Y_{k+1}^{(i)} = Y_k^{(i)} + \mu_i(Y_k, \theta) \Delta t_k + \sigma_{ij}(Y_k, \theta) \Delta B_k^{(j)} + \sigma_{ij}(Y_k, \theta) \frac{\partial \sigma_{ij}}{\partial y^{(i)}}(Y_k, \theta) \frac{1}{2} \left((\Delta B_k^{(j)})^2 - \Delta t_k \right) \quad (2)$$

for $k = 0, 1, \dots$ and where j is the column index of the one non-zero entry depending on $Y_k^{(i)}$ in the i^{th} row of the diffusion function.”

- p7 115 "denotes the multivariate Gaussian density with mean $a \in R^d$ and covariance matrix $b \in R^d \times d$ evaluated at y ." with my suggested addition for clarity.

was corrected

- p9 148 you claim that "n denotes the total number of data points in the augmented path" but it seems to me that, while there are n terms in the product in the displayed formula preceding this sentence, these n terms actually require X_{t_0} to X_{t_n} to compute, so that we have $n + 1$ data points in total.

We now write:

" $n + 1$ is the total number of data points in the augmented path"

- p20, Figure 8 : Green lines are hard to see here. Also, why is the posterior variance of alpha almost always underestimated? This is concerning in the application as it will lead to credible intervals that are too narrow and therefore deserves comment.

The colour was changed.

One cannot in general expect that the posterior distribution based on an approximation has a smaller variance than the true posterior distribution. But, of course, one has to be aware that the variance of the posterior distribution based on an approximation quantifies the uncertainty of the estimate with respect to the approximating model not the true model.

- p21 reads "One of the strengths of the original (Euler-based) MCMC scheme is its generic character and applicability. Through this, it possesses a practical advantage over otherwise more sophisticated methods such as the Exact Algorithm ([27]). This strength, however, is weakened by the limited applicability of the Milstein approximation especially in the multidimensional setting." I find this confusing. You are saying that the Euler-scheme's strength of generic character and applicability is weakened by the Milstein (not the Euler) approximation's limited applicability.

We rephrased the second sentence as follows:

"This strength does not translate to the Milstein-based MCMC scheme due to the limited applicability of the Milstein approximation especially in the multidimensional setting."

- p21 "Less iterations are completed for the methods involving the Milstein scheme and also the ESS is substantially lower." should probably read "Fewer iterations..."

was corrected

- Figures D1 and D2 have a similiar visibility problem with the dashed green lines.

The colour was changed.

Overall, apart from my one mathematical point querying whether the condition specified for all non-analytically computable double stochastic integrals to disappear is necessary, the content seems mathematically and algorithmically correct to me and I commend the authors for thoroughly and methodically investigating a natural open question and writing up this investigation properly even though the results do not lend themselves to making spectacular claims on improvements.